# ONE SAMPLE TO RULE THEM ALL:
# EXTREME DATA EFFICIENCY IN RL SCALING

## ABSTRACT

The reasoning ability of large language models (LLMs) can be unleashed with reinforcement learning (RL) (OpenAI, 2024; DeepSeek-AI et al., 2025a; Zeng et al., 2025). The success of existing RL attempts in LLMs usually relies on high-quality samples of thousands or beyond. In this paper, we challenge fundamental assumptions about data requirements in RL for LLMs by demonstrating the remarkable effectiveness of one-shot learning. Specifically, we introduce ***polymath learning***, a framework for designing one training sample that elicits multidisciplinary impact. We present three key findings: (1) A single, strategically selected math reasoning sample can produce significant performance improvements across multiple domains, including physics, chemistry, and biology with RL; (2) The math skills salient to reasoning suggest the characteristics of the optimal polymath sample; and (3) An engineered synthetic sample that integrates elements from multiple subjects outperforms training with individual samples that naturally occur. Our approach achieves superior performance to training with larger datasets across various reasoning benchmarks, demonstrating that sample quality and design, rather than quantity, may be the key to unlock enhanced reasoning capabilities in language models. Our results suggest a shift, dubbed as ***sample engineering***, toward precision engineering of training samples rather than simply increasing data volume.

## 1 INTRODUCTION

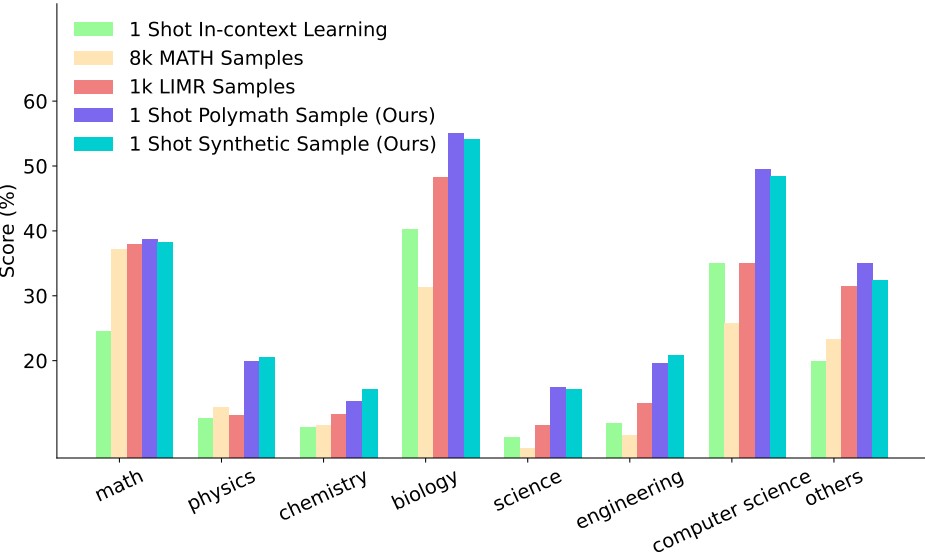

Figure 1: Reasoning capabilities of in-context learning, comprehensive learning (in MATH and LIMR) and polymath learning across different subject domains. The domain performance is averaged by subjects. We only mark the strongest in-context learning and polymath learning sample for demonstration purpose. Polymath learning in both natural sample and synthetic sample demonstrate significant gain over comprehensive learning in most domains.

Recent advances in Large Language Models (LLMs) have demonstrated the remarkable effectiveness of reinforcement learning (RL) in enhancing complex reasoning capabilities. Models like o1 (OpenAI, 2024), Deepseek R1 (DeepSeek-AI et al., 2025a), and Kimi1.5 (Team et al., 2025a) have shown that RL training is able to naturally induce sophisticated

reasoning behaviors, including self-verification (Weng et al., 2023), reflection (Shinn et al., 2023), and extended chains of thought. While these advances typically rely on large-scale training data, recent work has begun to challenge this paradigm. Li et al. (2025a) demonstrated with their LIMR approach that a strategically selected subset of just 1,389 samples can outperform the full 8k sample MATH dataset (Hendrycks et al., 2021). More recently, Wang et al. (2025a) made the surprising observation that even one single sample can produce meaningful improvements in math reasoning through RL, and Wang et al. (2025b) achieved similar gains by distilling high-quality reasoning paths from strong commercial models. However, this finding remains preliminary and math-specific, and leaves the critical questions of cross-domain generalization with internal abilities of LLMs unanswered: whether reasoning improvements beyond math can be achieved in similar manner? Whether a strategy exists in directing the optimal sample? Whether such sample can be synthesized to enhance the sample quality?

In this paper, we build upon these emerging insights to systematically investigate the phenomenon of one-shot reinforcement learning in broad reasoning tasks termed as *polymath learning*. Our central finding is that a single, carefully selected math reasoning sample is able to produce significant performance gains not only in mathematics but across diverse domains including physics, chemistry, biology, as well as more general reasoning domains. This cross-domain generalization suggests that RL may enhance fundamental reasoning mechanisms rather than merely domain-specific knowledge without saturated domain-specific training. Specifically, our work addresses three research questions:

**Cross-Domain Generalization:** Does a single mathematical reasoning sample yield improvements across diverse knowledge domains through polymath learning? We investigate the transfer mechanisms that allow fundamental reasoning patterns to transcend domain boundaries and observe that one single math sample selected on the math categories elicits greater reasoning gains of LLM than comprehensive datasets with thousands of samples, and the reasoning gains even extend to less quantitative subjects and domains that are distant from math.

**Optimal Sample Selection:** What characteristics define the ideal training sample for maximal impact in general reasoning domains? Although the optimal polymath sample varies across domains, we find that their efficacy correlates with the salient math skills critical to reasoning, particularly the skills in algebra and precalculus.

**Synthetic Sample Construction:** How can we engineer a hybrid "meta-sample" beyond naturally ocurred ones that integrates multiple reasoning skills? We propose a synthesis technique based on salient skill identification to construct the sample with comprehensive skills and multidisciplinary context. The results illustrate that the multidisciplinary background supports the comprehensiveness of the salient skills, and therefore benefits the cross-domain reasoning ability greater than the naturally-occurred samples that mainly possess math skills in limited categories and volumns. It shows the power of a single sample can be further amplified by properly integrating multidisciplinary knowledge.

By demonstrating that a single sample can trigger broad reasoning improvements, our findings adjust the understanding of data requirements in RL, suggesting that the field may benefit from a shift toward "***sample engineering***": the deliberate selection, and synthesis of training samples to unlock reasoning capabilities more efficiently, rather than simply scaling data volume and potentially induce generalization degration (Yang et al., 2024b).

## 2 RELATED WORK

**Reinforcement Learning in Language Models**   Reinforcement learning has been applied to aligning language models with human intents (Christiano et al., 2017) or instructions (Ouyang et al., 2022) through learning from human feedback. Later, it is extended to strengthen the long-form reasoning ability of models without relying on imitation of high-quality reasoning data, specifically by employing Reinforcement Learning with Verifierable Reward (RLVR) where the model outcomes can be verified and rewarded by verification functions with the advancement in RL algorithms (Schulman et al., 2017; Lambert et al., 2025; Hu et al., 2025a). However, training reliable outcome-based reward models (Cobbe et al., 2021) is challenging, and the rule-based reward function demonstrates effectiveness by simplifying the implementation of critic models and mitigating reward hacking (Shao et al., 2024). In this work, we extend the reasoning ability to broader reasoning domains by learning rewards from the mathematical problems.

**Data Efficiency in RL Training**   Xu et al. (2025) selects subset responses based on variance for GRPO training. And Zhang et al. (2025a) employs the most recent reward information for filtering prompts, which is shown to benefit GRPO training in Yu et al. (2025b). Other than focusing on the quality of prompt responses in RL training, Li et al. (2025a) highlights the significance of prompt quality by demonstrating the effectiveness of carefully selected training subset. Further, Shrestha et al. (2025) demonstrates cross-domain reasoning ability with less than 100 samples but requires a pre-warmup distillation stage, and Wang et al. (2025a) utilizes only one training sample and achieves a notable improvement in mathematical reasoning. And Zhao et al. (2025a) requires no human-expert data but still relies on an external executor to generate valid answers to synthetic coding problems. However, these studies still

focusing on the mathematical reasoning domain where the training data originates and neglect its broader impacts on multiple disciplines where the reasoning ability essences.

**Transfer Learning and Cross-Domain Generalization**    Afzal et al. (2024) demonstrates that small LLMs can catch up with larger counterparts in domain adaptation with few examples. And  Chen et al. (2024) adapts to new domain by extracting domain-invariant features in existing domain. Specifically for reasoning problems, Zhao et al. (2025a) unleashes improvement in mathematical reasoning soly based on training on programming data, and  Huan et al. (2025) demonstrates that RL achieves better generalization from math to other domains than supervised fine-tuning, without a deep dive into data efficiency. Li et al. (2025b) investigates the cross-domain impact in math reasoning, but only limits the study within logical-intensive domains like code and puzzle. In polymath learning, we enlarge the reasoning scope to various subjects and investigate the learning impact from one labeled mathematical sample.

**Sample Selection Strategies**    The effectiveness of finetuning large language models heavily relies on the quality of data selection (Xie et al., 2023). And well selected data samples can elicit powerful fine-tuning performance compared to data volume of magnitudes larger (Wang et al., 2023; Zhou et al., 2023). Xia et al. (2024) relies on the gradient information for data selection, while Liu et al. (2024b) formulates data selection as an optimal transportation problem. The effectiveness of data selection also extends to reasoning problems (Qin et al., 2024; Ye et al., 2025).  Liu et al. (2024a); Li et al. (2025c) apply LLM-based scores, justification, solve ratios (Havrilla et al., 2025) and LLM-based role-play (Luo et al., 2025a) to estimate sample diversity for data selection. Here we select polymath samples based on the alignment with reinforcement learning dynamics to elicit the reasoning ability in multiple disciplines. And we employ the salient-skill set to for selecting the synthesized data.

## 3  GRPO BASICS

Given a dataset $\mathcal{D} = \{(x, \hat{y})\}$ where $x$ and $\hat{y}$ stand for the prompt and golden answer, RLVR relies on a policy model $\pi_\theta(\cdot|x)$ to generate correct reasoning trajectories without relying on trajectories generated by human-expert or teacher models (Zhao et al., 2025a). In GRPO (Shao et al., 2024), the advantage value is estimated within a group of responses $G$ responses $\{y_1, y_2, ..., y_G\}$ to substitute the critic model in PPO while remaining effectiveness. Specifically,

$$\mathcal{L}_{GRPO} = \mathrm{E}_{[x \sim \mathcal{D}, \{y_i\} \sim \pi_{\theta_{\text{old}}}(\cdot|x)]} \Big[ \frac{1}{G} \sum_{i=1}^{G} \frac{1}{|y_i|} \sum_{t=1}^{|y_i|} \min(\tilde{r}_{i,t} A_i, \text{clip}(\tilde{r}_{i,t}, 1 - \epsilon, 1 + \epsilon) A_i) - \beta KL(\pi_\theta || \pi_{ref}) \Big]$$

$$A_i = \frac{r_i - \text{mean}(r_1, r_2, ...r_G)}{\text{std}(r_1, r_2, ...r_G)}, \quad \tilde{r}_{i,t} = \frac{\pi_\theta(y_{i,t}|x, y_{i,<t})}{\pi_{\theta_{\text{old}}}(y_{i,t}|x, y_{i,<t})}$$

Here $r_i$ is computed by applying the reward function on the response and the golden answer $r_i = \text{reward}(y_i, \hat{y}_i)$. $\pi_\theta(y_{i,t}|x, y_{i,<t})$ identifies the likelihood of the $t$-th token in $i$-th response from the policy model. Unlike previous efforts that assembles $\mathcal{D}$ with a comprehensive set of samples, in polymath learning, $\mathcal{D}_{polymath} = (x_1, \hat{y}_1)$.

## 4  POLYMATH LEARNING

 OpenAI et al. (2024) unlocks complex reasoning ability of LLM through reinforcement learning, and  DeepSeek-AI et al. (2025b;a) further demonstrates that such advanced reasoning ability can be elicited directly from pretrained base models using rule-based rewards, without relying on imitation from high-quality supervised reasoning trajectories. Existing explorations mainly focus on math or synthetic logic (Zeng et al., 2025; Pan et al., 2025; Xie et al., 2025) where large volumes of questions with rule-based verifiable answers are accessible. Beyond the success of *comprehensive learning*: training models with thousands of comprehensive high-quality problems and beyond, Wang et al. (2025a) shows that the reasoning ability can also be boosted by one single math sample with RL. Following this inquiry, we investigate *polymath learning*: training with one sample that plays a polymath role and extends the model reasoning power across domains. Similar to Wang et al. (2025a), we conduct polymath learning from math reasoning problems.

**Polymath Learning with One Natural Sample**    LIMR (Li et al., 2025a) displays the potential of improving training efficiency in reinforcement learning by selecting a subset of samples from MATH that closely align with the training dynamics of RL. A preliminary model is trained in LIMR to record the reward trajectories during optimization. The sample learnability is then computed by comparing its reward with the dataset-wise mean reward. Higher LIMR scores indicate greater alignment between the model behavior on individual sample and the entire dataset. However, learning from samples with excessively high LIMR scores risks over-specialization in math reasoning at the expense of broader

reasoning capabilities. Therefore, we select LIMR samples with the lowest scores (0.6) in different math categories as polymath candidates to maintain the same learnability according to preliminary experiments (See Appendix C for details). One polymath sample is displayed in Table 1 and others are included in Appendix N.

---

**Polymath Sample in Algebra**

[**Question**] A 100-gon $P_1$ is drawn in the Cartesian plane. The sum of the $x$-coordinates of the 100 vertices equals 2009. The midpoints of the sides of $P_1$ form a second 100-gon, $P_2$. Finally, the midpoints of the sides of $P_2$ form a third 100-gon, $P_3$. Find the sum of the $x$-coordinates of the vertices of $P_3$.

[**Answer**] 2009

---

Table 1: Polymath sample in algebra.

**Polymath Learning with One Synthetic Sample**  Synthesizing reasoning trajectories have been shown beneficial in boosting the reasoning ability in LLM in the pretraining (Ishibashi et al., 2025) and supervised-finetuning stage (Singh et al., 2024; Yuan et al., 2024). Careful problem synthesis also scales up the mathematical reasoning ability of models by reinforcement learning (Setlur et al., 2024). Since solving multidisciplinary problems and purely mathematical problems are not require on the same base of expertise, existing problem synthesis approaches based on problem imitation (Toshniwal et al., 2025), mutation (Havrilla et al., 2025) or creation based on seed concept or problem bank (Huang et al., 2025; Liang et al., 2025; Zhao et al., 2025b; Liu et al., 2025) do not directly apply. In practice, we find it challenging to organically integrate and align information from problems in diverse disciplines. Therefore, unlike Setlur et al. (2024) and Wang et al. (2025b), we synthesize the polymath sample based on instruction without relying on existing problems or models finetuned with question-generation (Ding et al., 2025; Wu et al., 2025b). Our final problem synthesis pipeline includes two stages,

- **Candidate problem generation** We employ strong models like OpenAI-O3 (OpenAI, 2025a), Gemini2.5-Pro (Google, 2025) and DeepSeek-R1 to include multidisciplinary knowledge from physics, chemistry and biology. The golden answers are collected from joint success of problem solving in those models.

- **Specialized problem selection** After massive generation of candidate problems, we employ Qwen2.5-72B-instruct to identify the salient math skills related in solving the problem given the problem text. The abundance of skills in different math categories is employed to reflect the complexities and qualities of problems. We then select the problems with the most specialized skills as the synthesized polymath samples, please refer to Appendix A for the prompt employed and Appendix K for example.

We find this instruction-based approach unleashes the creativity of LLMs in producing complex multidisciplinary problems. Specifically, we select the synthesized polymath sample with the most comprehensive skill spectrum (*Synthetic Prime*, shown in Table 2). Solving the *Synthetic Prime* requires a complex set of knowledge, including the strand sequence (biology), chemical bonds and energy to break bonds (chemistry), accumulating energy by collecting photons and estimating photon energy based on its wavelength (physics). The synthesis prompt is shown in Appendix A.

## 5 EXPERIMENTAL SETUP

We choose Qwen2.5-7b-base (Qwen et al., 2025) as the primary model, while Qwen2.5-math models (Yang et al., 2024a) demonstrate inferior performance on non-math benchmarks in preliminary experiments and are therefore not considered. Similar to Wang et al. (2025a), we employ GRPO (Shao et al., 2024) for RL training and augment the polymath sample into the batch of 128, and sample 16 responses per prompt with temperature of 1.0. The prompt template follows the design of Hu et al. (2025b). Following Huan et al. (2025), the model is trained for 140 steps since the reasoning ability saturates. We only employ a 0-1 outcome reward with rule-based matching of the final answer according to previous studies (Shao et al., 2024; Yu et al., 2025b), and exclude the format reward and the KL term as they demonstrate inferior performance (Wang et al., 2025a; Yu et al., 2025b). In skill identification, we employ *Algebra* to include salient skills from *Prealgebra*, *Algebra* and *Intermediate Algebra* to eliminate their large overlaps.

Our evaluation covers both math and non-math domains. Specifically, we select MATH500, AIME in 2024 and 2025, MinervaMath (Lewkowycz et al., 2022), GPQA-Diamond (Rein et al., 2024), Scibench (Wang et al., 2024a), MMLU-Pro (Wang et al., 2024b) with randomly select 100 problems for each subject and SuperGPQA (Team et al., 2025b) with 1500 random problems as the evaluation set. The full spectrum of subjects is listed in Appendix B. The model

---

**Polymath Sample (Synthetic Prime)**

[**Question**] A double-stranded DNA fragment of exactly 11 base pairs has the upper strand sequence 5 'G C G C G C G C A T A 3'.

Each adenine–thymine (A·T) base pair is held together by **2** hydrogen bonds, and each guanine–cytosine (G·C) base pair by **3** hydrogen bonds.
The DNA molecule is irradiated with monochromatic light of wavelength $\lambda = 400$nm. Assume that **100%** of every photon's energy is used exclusively to break hydrogen bonds between the two strands.
Use the exact data below (treat every value as exact):
* Enthalpy of one hydrogen bond $\Delta H = 20$kJ $\cdot$ mol$^{-1}$
* Planck constant $h = 6.626 10^{-34}$J $\cdot$ s
* Speed of light $c = 3.00 10^8$m$\cdot$ s$^{-1}$
* Avogadro constant $N_a = 6.022 10^{23}$mol$^{-1}$

**Fill in the blank:** What is the minimum number of 400 nm photons required to supply exactly enough energy to dissociate *all* hydrogen bonds in **one** molecule of this DNA fragment? (Answer with a single positive integer.)

[**Answer**] $\boxed{2}$

---

Table 2: The synthetic prime polymath sample that incorporates multidisciplinary knowledge.

responses are generated with greedy decoding in single attempt, except for AIME, where the results are averaged from 32 attempts with temperature being 0.4 (additional configurations are included in Appendix A).

## 6 RESULTS

### 6.1 CROSS-DOMAIN GENERALIZATION OF LEARNING ON SINGLE POLYMATH SAMPLE

Table 3 reports the reasoning performance aggregated by subject domains (e.g. *Math* includes all math problems from MATH500, AIME, MinervaMath and other benchmarks) by comparing model trained with different natural or synthetic polymath samples against the base model. Other than the *Synthetic Prime* sample, we also construct several synthetic specialist samples in different math categories by ranking the number of salient skills identified in these categories. We make several observations. Firstly, the base model exhibits skewed reasoning abilities: performing strong in math but weak in other domains. Secondly, polymath learning delivers substantial improvements over in-context learning across different subject domains. Thirdly, although comprehensive learning enhances the math reasoning ability, especially with effective data selection methods like LIMR, most natural polymath samples demonstrate comparable performance to comprehensive learning on the math domain, and surpass it on non-math domains (Figure 1), underscoring the potential of unleashing reasoning ability by one high-quality sample. Specifically, the polymath samples in prealgebra and precalculus stand out from all natural polymath samples, with their superior strength attributed to the broad coverage of salient math skills. Lastly, the synthetic polymath samples further elevate the reasoning ability. Most specialist samples outperform their natural polymath sample counterparts and demonstrate domain-specific advantages: geometry and algebra samples in engineering; number theory sample in math and probability sample in science. Furthermore, the *Synthetic Prime* sample achieves the strongest overall performance and demonstrates particular strength in physics and chemistry, suggesting that the reasoning potential of individual samples can be amplified through well-incorporation of multidisciplinary knowledge. Therefore we select the *Synthetic Prime* sample as the primary synthetic sample for the following experiments. Notably, unlike data collection approaches that are based on common-crawled data source (Wu et al., 2025a; He et al., 2025; Zhang et al., 2025b), we do not rely on seed data or observe evidence of data contamination in the polymath samples. The specialist samples are included in Appendix N.

We also breakdown the performance of N sampling (0-shot pass rate@64), polymath learning and comprehensive learning by subjects in Figure 2, with the subjects ordered by their similarities to math. The similarity is measured by computing the subject embedding distance between the mean of embeddings of all problems in each subject and the mean of problems in MATH500. We employ Text-Embedding-3-Small (OpenAI, 2025b) with the dimension of 1024 to generate problem representations. The best performance of polymath learning and in-context learning of polymath samples are displayed with triangles and stars respectively. We include our major findings as below

Table 3: The performance of employing different sample strategies on different subject domains. The best performance on each subject domain is bolded. Most natural polymath samples outperforms in-context learning and comprehensive learning with LIMR selection. Most synthetic specialist samples outperforms the corresponding natural sample, and the *Synthetic Prime* sample demonstrates the best performance. The dataset-wise results is included in Appendix C.

| Polymath Subject | Math | Physics | Chemistry | Biology | Science | Engineering | Computer Science | Others | Avg |
|---|---|---|---|---|---|---|---|---|---|
| **N=64 Sampling (0-shot)** | | | | | | | | | |
| - | 20.4 | 4.4 | 4.4 | 5.1 | 0.0 | 3.7 | 3.3 | 9.6 | 6.4 |
| **In-context Learning (1-shot)** | | | | | | | | | |
| **Natural Sample** | | | | | | | | | |
| Geometry | 24.5 | 8.0 | 7.2 | 24.4 | 4.3 | 6.0 | 29.0 | 11.6 | 14.4 |
| Prealgebra | 22.3 | 11.2 | 9.4 | 40.3 | 6.8 | 10.2 | 35.0 | 20.3 | 19.4 |
| Algebra | 21.4 | 10.9 | 9.8 | 38.7 | 8.3 | 10.4 | 35.0 | 20.6 | 19.4 |
| Intermediate Algebra | 22.7 | 8.0 | 7.0 | 21.8 | 4.5 | 9.5 | 32.0 | 15.5 | 15.1 |
| Number Theory | 21.7 | 10.9 | 8.7 | 31.9 | 5.4 | 6.6 | 28.0 | 15.8 | 16.1 |
| Precalculus | 21.6 | 8.3 | 5.9 | 20.2 | 5.2 | 6.8 | 26.0 | 11.9 | 13.2 |
| Probability | 22.4 | 9.7 | 7.2 | 24.4 | 5.6 | 7.7 | 22.0 | 13.2 | 14.0 |
| **Synthetic Sample** | | | | | | | | | |
| Prime | 18.6 | 4.6 | 4.6 | 8.4 | 2.2 | 4.6 | 11.0 | 7.7 | 7.7 |
| **Comprehensive Learning (> 1k shots)** | | | | | | | | | |
| **Natural Sample** | | | | | | | | | |
| MATH | 37.2 | 12.8 | 10.0 | 31.4 | 6.5 | 8.6 | 25.8 | 23.4 | 19.5 |
| LIMR | 38.0 | 11.6 | 11.8 | 48.3 | 10.0 | 13.4 | 35.1 | 31.5 | 25.0 |
| **Polymath Learning (1-shot) - Ours** | | | | | | | | | |
| **Natural Sample** | | | | | | | | | |
| Geometry | 15.5 | 9.9 | 10.0 | **55.1** | 11.2 | 16.7 | 37.1 | **35.0** | 23.8 |
| Prealgebra | 38.0 | 17.4 | 12.2 | 51.7 | 15.1 | 16.5 | **49.5** | 33.5 | 29.2 |
| Algebra | 37.3 | 17.4 | 13.7 | 51.7 | 12.1 | 15.6 | 43.3 | 30.9 | 27.7 |
| Intermediate Algebra | 36.3 | 19.1 | 13.1 | 50.0 | 13.9 | 17.5 | 42.3 | 31.1 | 27.9 |
| Number Theory | 37.7 | 16.9 | 12.4 | 49.2 | 13.4 | 17.8 | 42.3 | 32.2 | 27.7 |
| Precalculus | 38.0 | 18.4 | 13.7 | 50.0 | 16.0 | 19.7 | 43.3 | 31.0 | 28.8 |
| Probability | **38.8** | 19.9 | 11.5 | 46.6 | 14.7 | 16.4 | 41.2 | 31.4 | 27.6 |
| **Synthetic Sample** | | | | | | | | | |
| Geometry | 35.4 | 15.0 | 11.5 | 31.1 | 36.1 | **52.5** | 13.2 | 11.0 | 25.7 |
| Algebra | 37.3 | 16.9 | 12.6 | 31.5 | 41.2 | **52.5** | 18.6 | 13.9 | 28.1 |
| Number Theory | 38.4 | 18.2 | 12.0 | 32.1 | 36.1 | 47.5 | 18.6 | 13.8 | 27.1 |
| Precalculus | 37.1 | 20.3 | 15.3 | 32.9 | 44.3 | 48.3 | 20.8 | 16.5 | 29.4 |
| Probability | 37.1 | 16.7 | 13.9 | 30.1 | **46.4** | 50.0 | 19.7 | 10.8 | 28.1 |
| Prime | 38.3 | **20.6** | **15.7** | 54.2 | 15.6 | 20.8 | 48.5 | 32.4 | **30.8** |

**Strong mathematical but skewed reasoning of the base model**   Due to the massive mathematical and coding data participated in pretraining (Qwen et al., 2025; Wu et al., 2025a), the Qwen2.5-7b-base model achieves pass rate@64 > 0.5 in MATH500, higher than all other subjects with significant margins. However, the strength in MATH500 does not naturally extend to other subjects. For example, the base model performs poorly on physics, chemistry and biology, but demonstrates relative strength (pass rate@64 close to 0.2) in education, medicine, sociology and management, which does not possess similar proportion of quantitative components as math does.

**Comprehensive learning provides mathematical dominance, but not multidisciplinary**   Comprehensive learning with MATH or LIMR sets demonstrate strong performance in MATH500, and remain competitive with the strongest polymath sample in other math subjects (math, minerva). However, their performance on most non-math subjects lags behind by a large margin from the best polymath results. Their reasoning strengths gained from math-specific training only generalizes to a limited number of subjects, like economics, health, psychology, education, and history where more than fourfold performance improvement over zero-shot sampling is observed. Nonetheless, quality-driven data selection stays beneficial in comprehensive learning, with LIMR consistently outperforming MATH in most subjects.

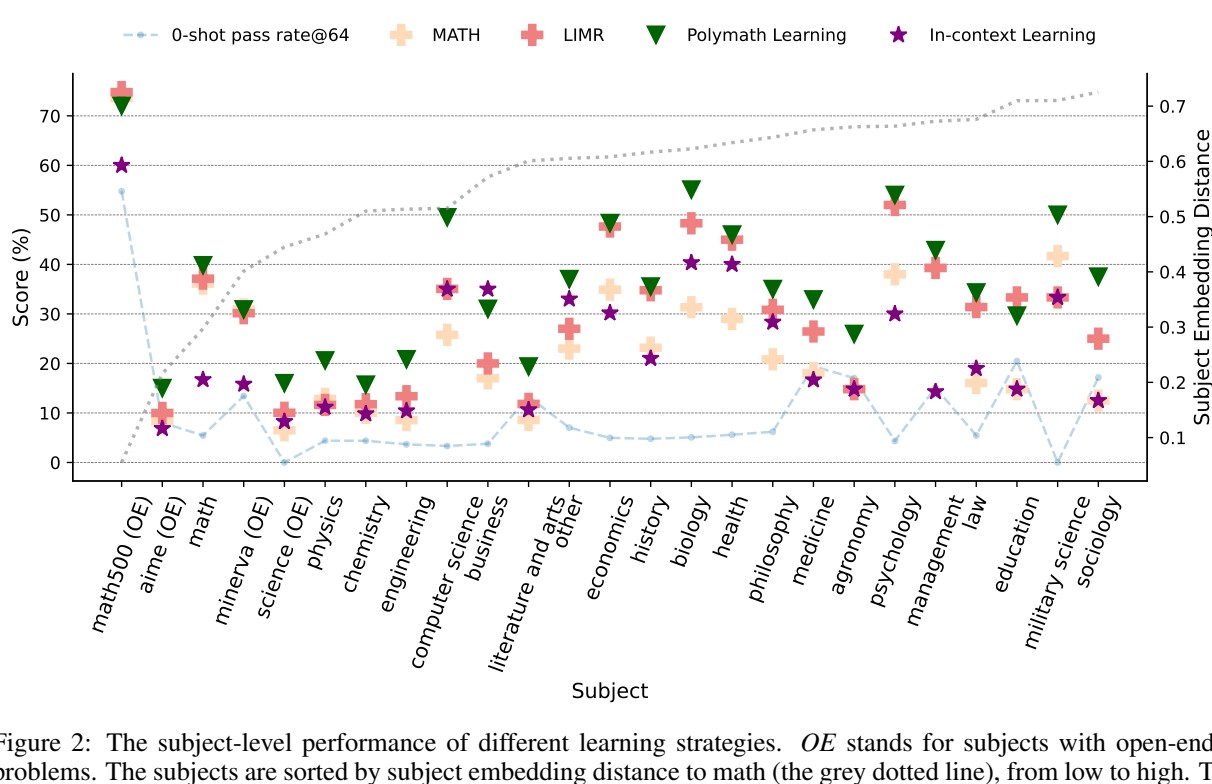

Figure 2: The subject-level performance of different learning strategies. *OE* stands for subjects with open-ended problems. The subjects are sorted by subject embedding distance to math (the grey dotted line), from low to high. The blue line represents pass ratio from 64 independent attempts of the base model. The stars and triangles represent best performance of in-context learning and polymath learning. Note that we only display the best polymath learning and in-context polymath learning results for demonstration.

**The effectiveness of in-context learning of polymath samples**   The best in-context polymath learning sample outperforms 0-shot pass rate@64 baseline in most subjects, indicating the benefits of polymath sample even in the gradient-free learning setting. Specifically, we find that the polymath sample in prealgebra or algebra under in-context learning demonstrate on-par or superior performance compared to at least one of comprehensive learning results in more than 50% subjects, with details included in Appendix J.

**Better generalization of polymath learning on math-distant subjects**   Even though the best polymath sample outperforms comprehensive learning in LIMR on subjects with heavy mathematical components like math and engineering. Its advantage is greater on subjects that are semantically distant from math, For example, around 10 points improvements in agronomy, literature and sociology. On average, polymath learning with the best natural samples achieves 14.5 points improvement on subjects with the 50% subjects farthest from MATH500 over comprehensive learning on the full MATH set, compared to 7.7 points on the 50% subjects that closet to MATH500, indicating that polymath learning confers stronger reasoning generalization on subjects that are semantically more math-distant.

## 6.2 CHARACTERISTICS OF OPTIMAL POLYMATH SAMPLE

Data diversity is beneficial in training more capable reasoning LLMs (Zhang et al., 2025b), serving both regularization to the neural network (Ba et al., 2025) and a mean to mitigate performance saturation especially when leveraging synthetic data sources (Jung et al., 2025). In polymath learning, we extend beyond the diversity at the level of problem or trajectory (Yu et al., 2025a)and instead examine the composition of salient mathematical skills within individual polymath samples. The result in Figure 3 illustrates the key supporting role of algebra and precalculus skills in cross-domain reasoning. Polymath samples demonstrating stronger performance tend to exhibit high prevalence of these skills. Furthermore, synthetic specialist samples with multidisciplinary backgrounds span a broader range of skills than math-specialized samples of the same speciality, which accounts for their superior performance. Notably, the *Synthetic Prime* sample exhibits the highest concentration of salient skills, suggesting that solving such problems requires a complex interplay of knowledge and thus provides rich learning signals for training LLMs.

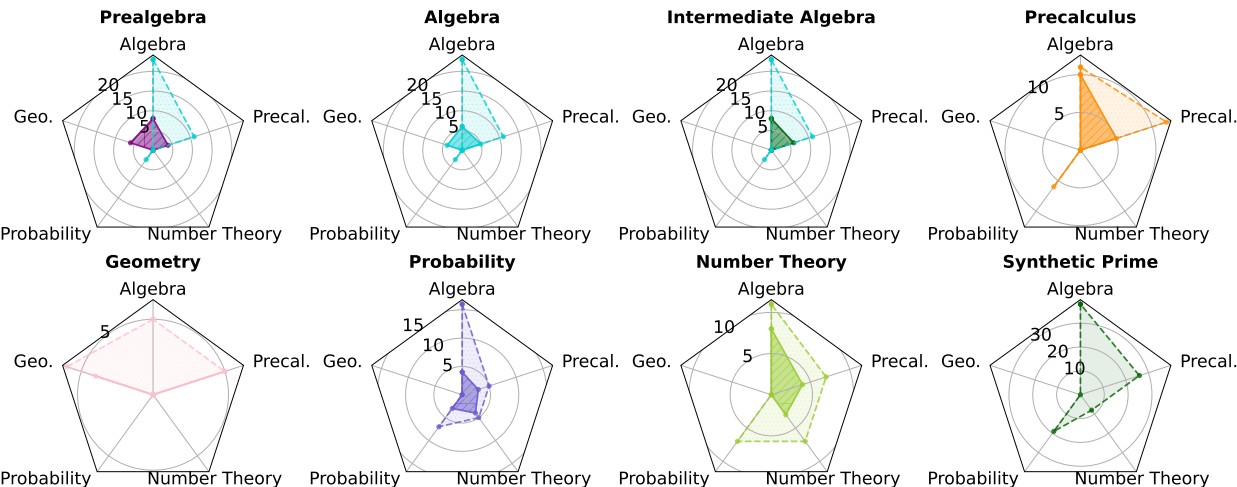

Figure 3: Skill spectrum between natural and synthetic polymath samples. The polygon represents number of salient skills identified in each math domain (*Geo.* and *Precal.* represents *Geometry* and *Precalculus* respectively). The real and dashed areas represent the natural and synthetic specialist samples. The last figure represents the *Synthetic Prime* sample, and the synthetic samples include more comprehensive salient skill sets than then natural polymath samples.

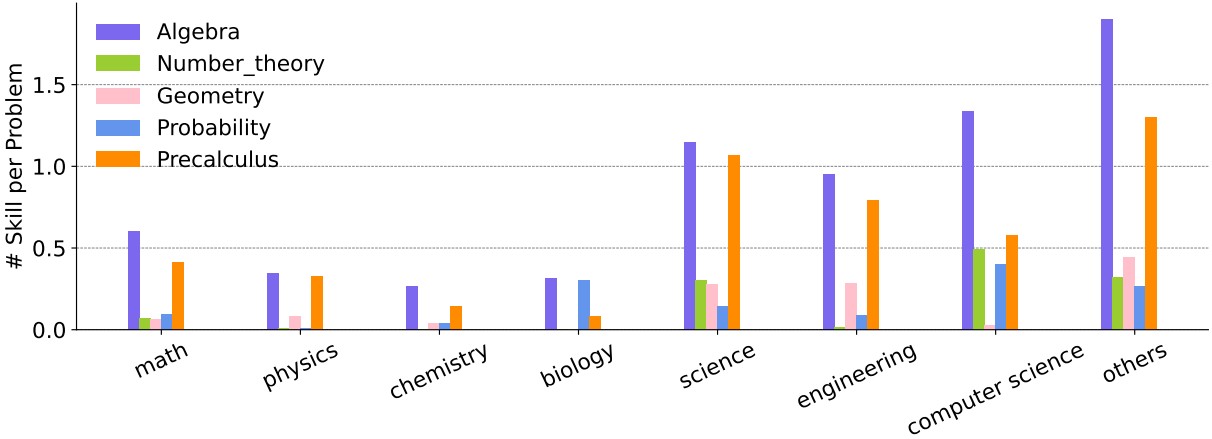

Figure 4: Average number of mathematical skills employed per problem in different subject domains. *Algebra* and *Precalculus* skills have the highest pupoluarities in all subject domains.

The distribution of salient skills across subject domains further highlights the central roles of algebra and precalculus. Skill abundance also reflects the degree of domain specialization. For instance, in engineering, the most frequent algebraic and geometric skills are *unit conversion* and *trigonometry*. Figure 4 shows that algebra and precalculus consistently dominate in skill popularity, underscoring their foundational importance for quantitative reasoning (e.g., *unit conversion* and *arithmetic operations*). Moreover, domains with integrative knowledge skills, such as science and engineering, demand more comprehensive combinations of salient skills compared to discipline-focused domains such as math, physics, chemistry, or biology.

# 7 GENERALIZATION OF SELF-VERIFICATION

The verification mechanism act as a signal for models to reconsider and refine their initial solutions (DeepSeek-AI et al., 2025a). Verification feedback can further enhance decision-making (Madaan et al., 2023; Shinn et al., 2023). To analyze such behavior, several signature words have been proposed for monitoring self-verification patterns (Xie et al., 2025). Following this, we collect pattern statistics across polymath learning samples, adding the 'code' category to capture python-based program verification and excluding 'reevaluate' for its rare appearance. We find that polymath

learning in general demonstrates more frequent self-verification behavior than comprehensive learning. Moreover, the polymath sample in 'number theory' and 'intermediate algebra' exhibit strong tendencies in eliciting the self-checking ('re-evaluate') behavior and programming assistance ('code') respectively. Moreover, different polymath samples display distinct self-verification preferences depending on the subject domain, with details in Appendix M.

Similar to Shao et al. (2025), we observe frequent use of program verification in the polymath sample of 'intermediate algebra'. However, the role of programs varies by domain: the programs in math are primarily used as part of the final answer generation process, including pseudo-execution errors like 'Timed out'; in physics and chemistry, by contrast, the programs are employed more for result validation. Importantly, without the access of external executor, the integration of program does not necessarily yield reasoning gains. Illustrative examples are provided in Appendix L.

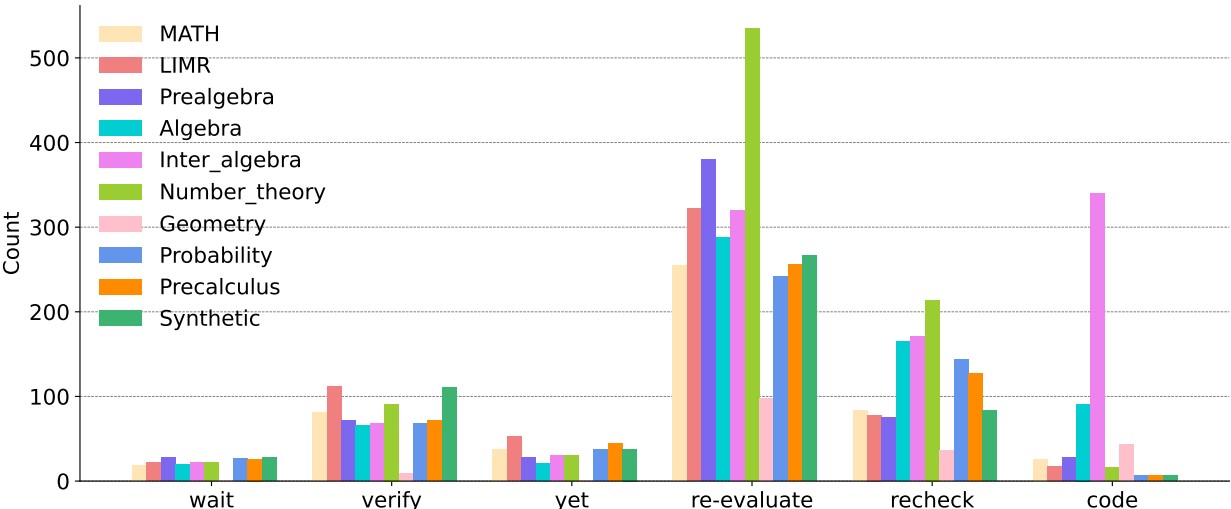

Figure 5: Self-verification patterns under different comprehensive and polymath samples across all subjects. Verification patterns like 're-evaluate' and 'recheck' appear most frequently in polymath learning with the 'number theory' sample, and the 'intermediate algebra' sample elicits the most code blocks in reasoning.

## 8 LIMITATIONS AND FUTURE WORK

In polymath learning, we focus our study in the effectiveness of one single training sample in lifting interdisciplinary reasoning ability with reinforcement learning. Due to resource constraints, our study only covers a small set of samples without larger-scale experiments in 1-shot polymath learning. And the skill-based selection does not extend to scaled skill-based problem synthesis like Havrilla et al. (2025). Although we observe different verification pattern preferences by choosing polymath samples, we do not observe direct connection between the self-verification and the improvement in reasoning abilities. Besides, the polymath learning experiments are only conducted in open-ended format, while previous studies has demonstrated the benefits of incorporating diverse question-answer formats (Akter et al., 2025), especially for benchmarks that are in multiple-choice formats. Moreover, our study is limited in polymath samples from math and does not extend to other domains where reliable rewards are accessible.

## 9 CONCLUSION

While math reasoning ability has been considered the primary metric to mark the progress of the reasoning of LLMs, the broader multidisciplinary reasoning abilities remain relatively underexplored. Inspired by the success of boosting math reasoning ability using one single training sample, we introduce polymath learning and show that training LLMs with one selected math sample can rival or even surpass datasets by orders of magnitude in eliciting reasoning across diverse domains. Our findings show that polymath learning yields stronger cross-domain reasoning ability than learning with the comprehensive math dataset, and sample synthesis further elevates the performance. Crucially, we trace this multidisciplinary reasoning potency of polymath samples to the abundance of salient math skills, especially in algebra and precalculus, within the reasoning structures of problems. Moreover, the synthesized samples with comprehensive salient skills tend to confer greater multidisciplinary reasoning strength, highlighting the promise of careful sample engineering as an alternative to indiscriminate data scaling.

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

# A  CONFIGURATIONS

We employ a learning rate of 1e-6 during training. And the maximum generation length is 2048. The configuration to collect zero-shot sampling for base model is listed in Table 4. The prompt used is displayed in Table 5, and the prompt to synthesize polymath samples is shown in Table 6. Around 500 candidate problems are synthesized on the candidate problem generation stage. The prompt employed for math skill identification is displayed in Table 7.

| HYPERPARAMETER | VALUE |
|:---:|:---|
| temperature | 0.5 |
| top k | 10 |
| top p | 0.8 |

Table 4: Hyperparameters for computing 0-shot pass rate@k of the base model.

---

**Prompt for Training**

A conversation between User and Assistant. The user asks a question, and the Assistant solves it. The assistant first thinks about the reasoning process in the mind and then provides the user with the answer. User: You must put your answer inside \\boxed{} and Your final answer will be extracted automatically by the \\boxed{} tag. For multiple choice questions, the final answer in \\boxed{} should be the option letter (A, B, C, D, etc.).
[PROBLEM]
Assistant:

---

Table 5: Training Prompt, where [PROBLEM] is the placeholder for the problem.

---

**Prompt for Synthesizing Polymath Sample**

You are a professor proficient in physics, chemistry, and biology, tasked with creating a highly integrated problem for students that encompasses knowledge from all three disciplines. This problem should be a fill-in-the-blank question, and the final answer must be a precise integer (a positive integer between 1-1000). The difficulty of this question should be at the high school to university level. Furthermore, it should not involve any estimation, and complex calculations should be avoided as much as possible to ensure the robustness of the evaluation.

---

Table 6: Prompt for synthesizing polymath sample.

---

**Prompt for Skill Identification**

Here is a reasoning problem, and your job is to identify the concepts and skills in the scope of [CATEGORY] that are related to solve the problem.
Please separate the concepts or skills with ;, and if there is no skills or concepts identified, please answer with None. Please put your answer within <answer></answer>.
For example: compute derivatives is the skill in precalculus.
Question:
[QUESTION]

---

Table 7: Prompt for skill identification. The [CATEGORY] and [QUESTION] are the placeholder for math category (e.g. algebra) and problem respectively.

# B  FULL SUBJECT LIST

The full list of reasoning subjects being evaluated is displayed in Table 8.

## C  RESULTS BY DATASETS

Table 9 includes results by datasets on polymath learning and comprehensive learning, with the synthetic sample still performing the strongest.

| SUBJECT DOMAIN | SUBJECT | SOURCE | # SAMPLES |
|---|---|---|---|
| Math | AIME | AIME2024, AIME2025 | 60 |
| | MATH500 | MATH | 500 |
| | Minerva | MinervaMath | 272 |
| | math | Scibench, MMLU-Pro | 299 |
| Physics | physics | GPQA-Diamond, Scibench, MMLU-Pro | 413 |
| Chemistry | chemistry | GPQA-Diamond, Scibench, MMLU-Pro | 459 |
| Biology | biology | GPQA-Diamond, Scibench, MMLU-Pro | 118 |
| Science | science | SuperGPQA | 557 |
| Engineering | engineering | SuperGPQA | 447 |
| Computer Science | computer science | MMLU-Pro | 100 |
| Others | military science | SuperGPQA | 12 |
| | business | MMLU-Pro | 100 |
| | philosophy | MMLU-Pro, SuperGPQA | 120 |
| | economics | MMLU-Pro, SuperGPQA | 149 |
| | management | SuperGPQA | 28 |
| | health | MMLU-Pro | 100 |
| | psychology | MMLU-Pro | 100 |
| | medicine | SuperGPQA | 155 |
| | education | SuperGPQA | 27 |
| | agronomy | SuperGPQA | 27 |
| | literature and arts | SuperGPQA | 93 |
| | law | MMLU-Pro, SuperGPQA | 137 |
| | history | MMLU-Pro, SuperGPQA | 138 |
| | sociology | SuperGPQA | 8 |
| | other | MMLU-Pro | 100 |

Table 8: Evaluation reasoning benchmarks with subjects included.

## D  LIMR SCORE BASICS

The LIMR score Li et al. (2025a) is computed by measuring the sample-wise training reward with the dataset-wise average. Specifically,

$$s_i = 1 - \frac{\sum_{i=1}^{K}(r_i^k - \bar{r}^k)^2}{\sum_{i=1}^{K}(1 - \bar{r}^k)^2}, \quad \bar{r}^k = \frac{1}{N}\sum_{i=1}^{N} r_i^k$$

where $r_i^k$ is the reward of sample $i$ in the $k$-th epoch, and $\bar{r}^k$ is the average reward of training set in the $k$-th epoch.

## E  SAMPLE PREFERENCE WITH LIMR SCORES

We include the results from selecting different LIMR scores from two math categories, *prealgebra* and *probability*, that demonstrate strong multidisciplinary reasoning ability. The results in Figure 6 show that the samples with LIMR score equals 0.6 perform best.

Table 9: Results on different reasoning benchmarks, where *OE* refers to benchmarks of open-ended problems: MATH500, AIME2024, AIME2025, Minerva and Scibench, while *MCQ* refers to benchmarks of multiplechoice problems. The best performance is bolded and the best polymath learning performance is underlined if not optimal.

| Polymath Subject | MATH500 | AIME2024 | AIME2025 | Minerva | GPQA-Diamond | SuperGPQA | MMLU-Pro | SciBench | AVG-OE | AVG-MCQ | AVG-All |
|---|---|---|---|---|---|---|---|---|---|---|---|
| **N=64 Sampling (0 shot)** | | | | | | | | | | | |
| - | 54.8 | 9.0 | 7.1 | 13.4 | 13.1 | 15.7 | 4.7 | 9.8 | 23.6 | 11.3 | 15.9 |
| **In-context Learning (1 shot)** | | | | | | | | | | | |
| **Natural Sample** | | | | | | | | | | | |
| Geometry | 60.0 | 8.2 | 4.7 | 15.4 | 9.6 | 4.5 | 20.5 | 6.8 | 19.0 | 11.5 | 16.2 |
| Prealgebra | 55.0 | 9.2 | 4.5 | 10.7 | 16.2 | 9.2 | 28.8 | 6.4 | 17.2 | 18.1 | 17.5 |
| Algebra | 48.0 | 8.2 | 3.1 | 15.8 | 14.6 | 10.7 | 25.6 | 6.7 | 16.4 | 17.0 | 16.6 |
| Intermediate Algebra | 59.6 | 5.1 | 4.5 | 12.1 | 14.1 | 7.3 | 20.5 | 5.7 | 17.4 | 14.0 | 16.1 |
| Number Theory | 52.8 | 8.5 | 3.9 | 11.8 | 16.7 | 6.3 | 23.4 | 5.9 | 16.6 | 15.5 | 16.2 |
| Precalculus | 51.8 | 6.7 | 3.9 | 15.8 | 13.1 | 4.9 | 19.0 | 5.2 | 16.7 | 12.3 | 15.0 |
| Probability | 54.2 | 7.3 | 4.0 | 13.6 | 11.1 | 6.3 | 19.7 | 5.8 | 17.0 | 12.4 | 15.2 |
| **Synthetic Sample** | | | | | | | | | | | |
| Synthetic | 44.2 | 4.8 | 2.4 | 15.1 | 5.6 | 2.8 | 10.6 | 3.8 | 14.1 | 6.3 | 11.2 |
| **Comprehensive Learning (> 1k shots)** | | | | | | | | | | | |
| **Natural Sample** | | | | | | | | | | | |
| MATH (8k) | 73.6 | 13.0 | 7.9 | **30.9** | 11.7 | 10.3 | 22.5 | **23.1** | 29.7 | 14.8 | 24.1 |
| LIMR (1k) | **74.8** | 12.6 | 8.9 | 30.1 | 13.2 | 15.8 | 31.5 | 22.7 | 29.8 | 20.2 | 26.2 |
| **Polymath Learning (1 shot)** | | | | | | | | | | | |
| **Natural Sample** | | | | | | | | | | | |
| Geometry | 26.6 | 0.0 | 0.0 | 19.9 | **23.9** | 18.5 | 33.1 | 7.9 | 10.9 | 25.2 | 16.2 |
| Prealgebra | 71.2 | 13.3 | 13.3 | 30.9 | 18.3 | 19.4 | 35.0 | 21.4 | 30.0 | 24.2 | **27.9** |
| Algebra | 72.0 | 6.7 | 0.0 | **30.9** | 16.2 | 17.3 | 34.9 | 22.8 | 26.5 | 22.8 | 25.1 |
| Intermediate Algebra | 71.2 | 13.3 | 0.0 | 28.7 | 20.3 | 18.9 | 34.5 | 22.0 | 27.0 | 24.6 | 26.1 |
| Number Theory | 69.6 | **16.7** | 10.0 | **30.9** | 17.8 | 18.2 | 35.0 | 22.3 | 29.9 | 23.7 | 27.6 |
| Precalculus | 71.6 | 10.0 | 10.0 | 30.5 | 18.8 | 20.9 | 34.1 | 22.4 | 28.9 | 24.6 | 27.3 |
| Probability | 71.6 | 13.3 | **16.7** | 29.8 | 14.2 | 18.9 | 34.9 | 22.7 | **30.8** | 22.7 | **27.8** |
| **Synthetic Sample** | | | | | | | | | | | |
| Geometry | 71.4 | 10.2 | 6.7 | 27.2 | 15.7 | 16.9 | 30.7 | 21.4 | 27.4 | 21.1 | 25.0 |
| Algebra | 71.6 | 10.2 | 6.7 | **30.9** | 20.3 | 19.3 | 33.6 | 21.8 | 28.2 | 24.4 | 26.8 |
| Number Theory | 73.8 | 11.7 | 7.1 | 29.8 | 14.2 | 19.3 | 34.6 | **23.1** | 29.1 | 22.7 | 26.7 |
| Precalculus | 71.8 | 11.4 | 7.7 | 29.4 | 19.8 | **21.5** | 35.8 | 22.8 | 28.6 | 25.7 | 27.5 |
| Probability | 71.8 | 11.6 | 7.2 | 28.3 | 16.8 | 17.5 | 36.4 | 22.1 | 28.2 | 23.6 | 26.5 |
| Prime | 71.4 | 10.1 | 7.2 | **30.9** | 21.3 | 20.5 | **38.4** | 22.3 | 28.4 | **26.7** | **27.8** |

---

### Skill Identification Sample - Science

**[Question]** A particle of mass 1 kg is moving in the $x - y$ plane and its potential energy $U$ in joule obeys the law $U = 6x + 8y$, where $(x, y)$ are the coordinates of the particle in meter. If the particle starts from rest at (9,3) at time $t = 0$, then

(A): The speed of the particle when it crosses the y axis is $5\sqrt{3} m/s$

(B): The speed of the particle when it crosses y axes is $7\sqrt{3} m/s$

(C): Magnitude of acceleration of particle is $10 m/s^2$

(D): The speed of the particle when it crosses y axes is $11\sqrt{3} m/s$

(E): Acceleration of particle is zero

(F): The speed of the particle when it crosses y axes is $8\sqrt{3} m/s$

(G): The speed of the particle when it crosses y axes is $9\sqrt{3} m/s$

(H): The speed of the particle when it crosses y axes is $12\sqrt{3} m/s$

(I): The speed of the particle when it crosses y axes is $10\sqrt{3} m/s$

**[Skills in Algebra]**

Interpreting physical laws in mathematical form

Understanding the relationship between potential energy and force

Using the gradient to find force components

Applying the work-energy theorem

Solving for velocity using energy conservation

Understanding the relationship between force and acceleration

Solving for acceleration using newton's second law

Analyzing motion in two dimensions

Solving for the time when a particle crosses a specific axis

Evaluating expressions involving square roots

Table 10: Skills identified from a sample science problem. Salient skills in other math categories are not identified.

## F    ROBUSTNESS OF THE RESULTS

We include the results of comprehensive learning in MATH dataset and polymath learning in the synthetic prime sample with 3 independent runs on Qwen2.5-7b-base. The results in Table 11 shows that the comprehensive learning

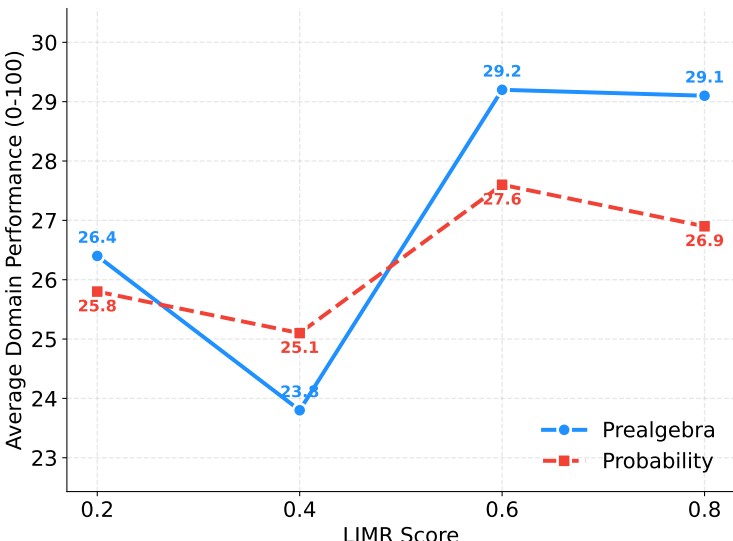

Figure 6: Average domain performance over samples of different LIMR scores. The performance is reported the same way as in Table 3. The samples with LIMR score being 0.6 outperform others.

on 8k MATH samples demonstrate stronger reasoning in math benchmarks, but the synthetic prime sample outperforms the 8k MATH training set in most other benchmarks as well as the average performance.

Table 11: The results of comprehensive learning on MATH and polymath learning on synthetic prime sample with 3 independent runs in Qwen2.5-7b-base. Polymath learning with the synthetic prime sample outperforms on most benchmarks as well as the overall performance.

| Polymath Subject | MATH500 | AIME2024 | AIME2025 | Minerva | GPQA-Diamond | SuperGPQA | MMLU-Pro | SciBench | AVG-OE | AVG-MCQ | AVG-All |
|---|---|---|---|---|---|---|---|---|---|---|---|
| Comprehensive Learning (> 1k shots) | | | | | | | | | | | |
| MATH (8k) | 73.0±0.59 | 15.6±4.16 | 6.7±0.0 | 29.5±1.24 | 11.9±0.24 | 11.6±1.75 | 25.0±2.94 | 23.5±0.37 | 29.7±0.73 | 16.2±1.53 | 24.6±0.72 |
| Polymath Learning (1 shot) | | | | | | | | | | | |
| Prime | 71.7±0.34 | 12.2±1.56 | 10.0±4.71 | 31.0±1.07 | 20.3±0.71 | 20.8±0.31 | 38.1±0.69 | 21.9±0.33 | 29.4±1.03 | 26.4±0.29 | 28.2±0.62 |

# G  PERFORMANCE ON MMLU-PRO AND SUPERGPQA

The results on the full set of MMLU-Pro and SuperGPQA of comprehensive learning in 8k MATH samples and polymath learning in the synthetic prime sample trained with Qwen2.5-7b-base and greedy decoding are included in Table 12. Polymath learning in the synthetic prime sample significantly outperforms both 0-shot and comprehensive learning in 8k MATH samples.

Table 12: Performance of comprehensive learning in the 8k MATH samples and polymath learning in the synthetic prime sample on the full set of MMLU-Pro and SuperGPQA, the synthetic prime sample performs best.

| Data | MMLU-Pro | SuperGPQA |
|---|---|---|
| 0-shot | 30.3 | 16.8 |
| MATH (8k) | 31.7 | 16.6 |
| Prime | **37.6** | **21.7** |

## H    TRAINING DYNAMICS OF POLYMATH LEARNING

Figure 7 illustrates the training dynamics of comprehensive learning and polymath learning across natural and synthetic samples. We specifically prolong the training on the 8k MATH training set to better observe convergence. We observe that comprehensive learning, on either the 8k MATH training set or the LIMR subset, yields progressive improvement on MATH500, but exhibits pronounced overfitting on multidisciplinary benchmarks such as GPQA Diamond, SuperGPQA, and MMLU-Pro. And training with the MATH set exacerbates this effect. Polymath learning, on the other hand, demonstrates substantially greater robustness on multidisciplinary reasoning benchmarks. Moreover, both the synthetic prime sample and natural polymath sample in prealgebra deliver stronger multidisciplinary reasoning performance than the $pi_1$ employed in prior works (Wang et al., 2025a;b), which is selected from a dataset more challenging than MATH.

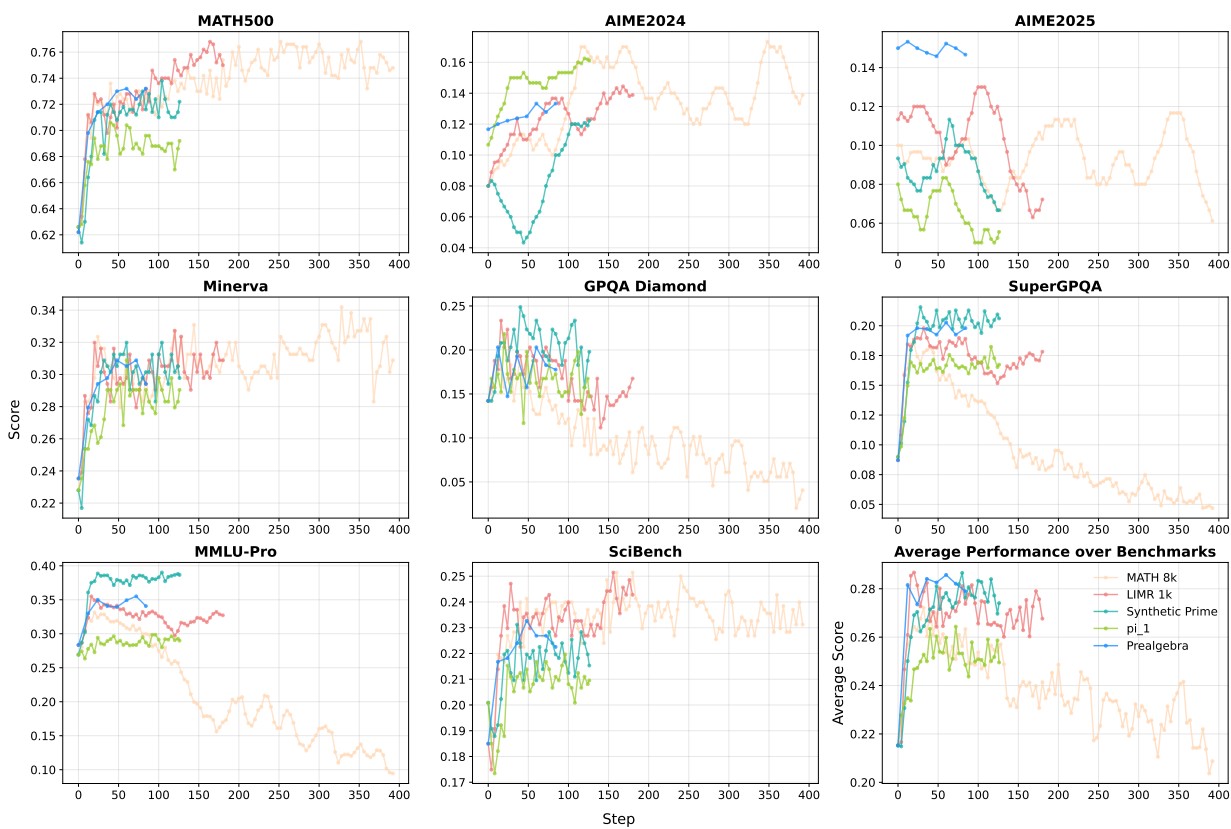

Figure 7: The evaluation results of benchmarks between comprehensive learning and different polymath learning samples (synthetic prime sample, natural prealgebra sample, $pi_1$) trained in Qwen2.5-7b-base. The results are collected in greedy decoding and rolling average is applied to AIME2024, AIME2025 for demonstration purpose.

## I    POLYMATH LEARNING WITH OTHER 1-SHOT SAMPLE

Previous success in reinforcement learning with one example (Wang et al., 2025a;b) selects $\pi_1$ from DeepScaleR (Luo et al., 2025b) (see Table 14), a curated dataset of challenging mathematical competition problems. Results in Table 13 demonstrate the effectiveness of synthetic prime sample over both $\pi_1$ and comprehensive learning with 8k MATH samples in both Qwen2.5-7b-base and Qwen2.5-14b-base.

## J    REASONING BREAKDOWN BY SUBJECT

Figure 8 illustrates the best polymath sample for different subjects.

Table 13: The results between comprehensive learning on 8k MATH samples and polymath learning on the synthetic prime sample and $\pi_1$ in Qwen2.5-7b-base and Qwen2.5-14b-base. The synthetic prime sample consistently outperforms the other two data choices across models.

| Data | Math | Physics | Chemistry | Biology | Science | Engineering | Computer Science | Others | Avg |
|---|---|---|---|---|---|---|---|---|---|
| Qwen2.5-7b-base | | | | | | | | | |
| N=64 Sampling (0-shot) | | | | | | | | | |
| - | 20.4 | 4.4 | 4.4 | 5.1 | 0.0 | 3.7 | 3.3 | 9.6 | 6.4 |
| Comprehensive Learning ($>$ 1k shots) | | | | | | | | | |
| MATH | 37.2 | 12.8 | 10.0 | 31.4 | 6.5 | 8.6 | 25.8 | 23.4 | 19.5 |
| Polymath Learning (1-shot) | | | | | | | | | |
| $\pi_1$ (DeepScaleR) | 35.5 | 14.3 | 11.3 | 28.4 | **35.1** | **44.1** | 13.8 | 10.4 | 24.1 |
| Prime | **38.3** | **20.6** | **15.7** | **54.2** | 15.6 | 20.8 | **48.5** | **32.4** | **30.8** |
| Qwen2.5-14b-base | | | | | | | | | |
| N=64 Sampling (0-shot) | | | | | | | | | |
| - | 37.7 | 26.2 | 22.2 | 28.1 | 41.2 | 39.0 | 20.8 | 14.3 | 28.7 |
| Comprehensive Learning ($>$ 1k shots) | | | | | | | | | |
| MATH | 42.7 | 26.4 | 20.5 | **44.7** | 49.5 | **64.4** | 22.3 | 15.6 | 35.8 |
| Polymath Learning (1-shot) | | | | | | | | | |
| $\pi_1$ (DeepScaleR) | 40.4 | 27.6 | 20.0 | 39.4 | 51.5 | 57.6 | 22.1 | 17.1 | 34.5 |
| Prime | **44.0** | **32.7** | **22.7** | 42.3 | **56.7** | 58.5 | **31.0** | **20.6** | **38.6** |

---

**The $\pi_1$ Sample**

[**Question**] The pressure P exerted by wind on a sail varies jointly as the area A of the sail and the cube of the wind's velocity V. When the velocity is 8 miles per hour, the pressure on a sail of 2 square feet is 4 pounds. Find the wind velocity when the pressure on 4 square feet of sail is 32 pounds.

[**Answer**] 12.8

---

Table 14: The $\pi_1$ sample.

## K  EXAMPLE OF SALIENT MATH SKILL IN THE REASONING PROBLEM

A sample science problem and relevant algebra skills to solve is displayed in Table 10.

## L  SELF-VERIFICATION EXAMPLES

Table 15, Table 16 and Table 17 include examples in math, physics, and chemistry problems where program verification emerges in polymath learning with the polymath sample in 'intermediate algebra'.

## M  SELF-VERIFICATION BY SUBJECT DOMAINS

We list the self-verification statistics by different sbuject domains in Figure 9 and Figure 10. Specifically, we found that 'verify' is more preferred in math problems while 're-evaluate' is appeared more frequently in science and engineering problems. Besides, polymath learning with the 'intermediate algebra' sample elicits the most coding verification among all the polymath and comprehensive samples.

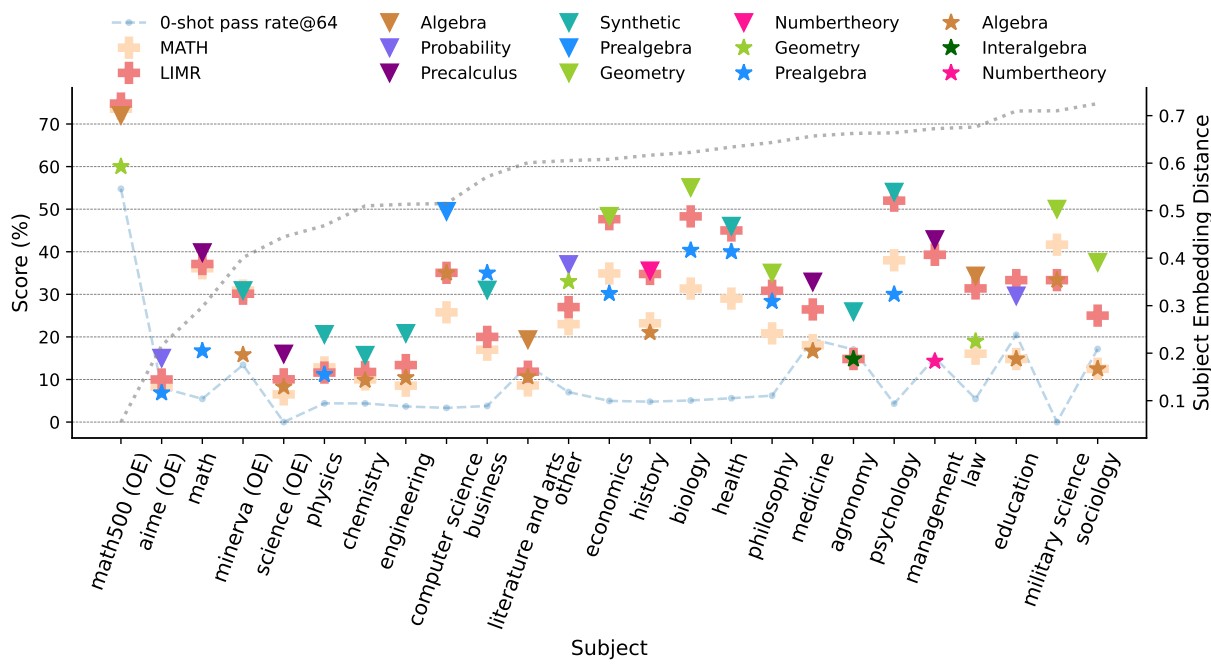

Figure 8: The subject-level performance of different learning strategies. *OE* stands for subjects with open-ended problems. The subjects are sorted by subject embedding distance to math (the grey dotted line), from low to high. The blue line represents pass ratio from 64 independent attempts of the base model. The stars and triangles represent best performance of in-context learning and polymath learning. Note that we only display the best polymath learning and in-context polymath learning results for demonstration, and *Synthetic* represents the *Synthetic Prime* sample.

---

**Self Verification - Math**

**[Question]** The intensity of light with wavelength $\lambda$ traveling through a diffraction grating with $N$ slits at an angle $\theta$ is given by $I(\theta) = N^2 \sin^2 k / k^2$, where $k = (\pi N d \sin \theta)/\lambda$ and $d$ is the distance between adjacent slits. A helium-neon laser with wavelength $\lambda = 632.8 \times 10^{-9}$ m is emitting a narrow band of light, given by $-10^{-6} < \theta < 10^{-6}$, through a grating with 10,000 slits spaced $10^{-4}$ m apart. Use the Midpoint Rule with $n = 10$ to estimate the total light intensity $\int_{-10^{-6}}^{10^{-6}} I(\theta)d\theta$ emerging from the grating.

**[Answer]** $\boxed{59.4}$

**[Intermediate Algebra]** To solve this problem, we follow these steps:

1. Define the function $I(\theta) = N^2 \sin^2 \left( \frac{\pi N d \sin \theta}{\lambda} \right) / \left( \frac{\pi N d \sin \theta}{\lambda} \right)^2$.

2. Substitute the given values $N = 10000$, $d = 10^{-4}$, and $\lambda = 632.8 \times 10^{-9}$.

3. Use the Midpoint Rule with $n = 10$ to approximate the integral $\int_{-10^{-6}}^{10^{-6}} I(\theta)\, d\theta$. Let's break it down into manageable parts and implement it in Python using the sympy library.

```python
import sympy as sp
# Define the variables
theta = sp.symbols('theta')
# Given values
N = 10000
d = 10**-4
lambda_val = 632.8 * 10**-9
# Define the function I(theta)
k = (sp.pi * N * d * sp.sin(theta)) / lambda_val
I_theta = N**2 * sp.sin(k)**2 / k**2
# Define the limits of integration
a = -10**-6
b = 10**-6
n = 10
# Midpoint Rule
delta_theta = (b - a) / n
midpoints = [a + (i + 0.5) * delta_theta for i in range(n)]
integral_approx = sum(I_theta.subs(theta, midpoint) for midpoint in midpoints) * delta_theta
# Evaluate the integral approximation
integral_approx_evaluated = integral_approx.evalf()
print(integral_approx_evaluated)
```

`Output: TimeoutError:` `'Timed Out'` It seems that the computation took too long to complete. This might be due to the complexity of the function and the large number of terms involved in the Midpoint Rule. Let's simplify the problem by breaking it down further and using numerical integration instead of symbolic integration.

We can use the 'scipy.integrate.quad' function to perform the numerical integration. This function is designed to handle such integrals efficiently.

Here's the updated code using numerical integration:

```python
import numpy as np
from scipy.integrate import quad
# Given values
N = 10000
d = 10**-4
lambda_val = 632.8 * 10**-9
# Define the function I(theta)
def I_theta(theta):
    k = (np.pi * N * d * np.sin(theta)) / lambda_val
    return N**2 * np.sin(k)**2 / k**2
# Define the limits of integration
$a = -10**-6$
$b = 10**-6$
# Perform the numerical integration
integral_approx, error = quad(I_theta, a, b)
# Print the result
print(integral_approx)
```

`Output: 10000000000.0` The numerical integration result for the total light intensity $(\int_{-10^{-6}}^{10^{-6}} I(\theta)d\theta)$ emerging from the grating is $\boxed{10000000000.0}$.

Table 15

---

**Self Verification - Physics**

[Question] We know that the negative charge on the electron and the positive charge on the proton are equal. Suppose, however, that these magnitudes differ from each other by $0.00010\%$. With what force would two copper coins, placed $1.0$ m apart, repel each other? Assume that each coin contains $3 \times 10^{22}$ copper atoms. (Hint: A neutral copper atom contains 29 protons and 29 electrons.)

[Answer] 1.7

[Intermediate Algebra] To solve this problem, we need to follow these steps:
1. Calculate the charge difference between the electron and the proton.
2. Determine the total charge on each copper coin.
3. Use Coulomb's law to find the force between the two copper coins.

Let's break it down:
1. **Calculate the charge difference:**
- The charge of an electron is $e = -1.602 \times 10^{-19}$ C.
- The charge of a proton is $e = 1.602 \times 10^{-19}$ C.
- The difference in charge is $\Delta e = 0.00010\% \times 1.602 \times 10^{-19}$ C.
- $0.00010\% = 0.000001$, so $\Delta e = 0.000001 \times 1.602 \times 10^{-19} = 1.602 \times 10^{-25}$ C.

2. **Determine the total charge on each copper coin:**
- Each copper atom has 29 protons and 29 electrons, so the net charge of a neutral copper atom is zero.
- Each copper coin contains $3 \times 10^{22}$ copper atoms, so the total charge on each coin is $3 \times 10^{22} \times 1.602 \times 10^{-25} = 4.806 \times 10^{-3}$ C.

3. **Use Coulomb's law to find the force between the two copper coins:**
- Coulomb's law states that the force $F$ between two point charges $q_1$ and $q_2$ separated by a distance $r$ is given by $F = k\frac{q_1 q_2}{r^2}$, where $k$ is the Coulomb constant $8.99 \times 10^9 \ \frac{N \cdot m^2}{C^2}$.
- Here, $q_1 = q_2 = 4.806 \times 10^{-3}$ C and $r = 1.0$ m.
- So, $F = 8.99 \times 10^9 \times \frac{(4.806 \times 10^{-3})^2}{1.0^2} = 8.99 \times 10^9 \times 2.31 \times 10^{-5} = 2.07 \times 10^5$ N.

Let's implement this in Python to ensure the accuracy of the result:

```python
# Constants
e = 1.602e-19  # charge of an electron (and proton)
delta_e = 0.00010 / 100 * e  # charge difference
num_atoms = 3e22  # number of copper atoms in each coin
charge_per_atom = e  # charge of a single copper atom
total_charge_per_coin = num_atoms * charge_per_atom  # total charge on each coin
k = 8.99e9  # Coulomb constant
distance = 1.0  # distance between the coins in meters

# Calculate the force using Coulomb's law
force = k * (total_charge_per_coin ** 2) / (distance ** 2)
print(force)
```

output:   206673.0

The force between the two copper coins, each containing $3 \times 10^{22}$ copper atoms, placed $1.0$ meter apart, with a charge difference of $0.00010\%$ between the electron and the proton, is $2.07 \times 10^5$ N .

Table 16

**Self Verification - Chemistry**

[**Question**] Assume all gases are perfect unless stated otherwise. Note that 1 atm = 1.013 25 bar. Unless otherwise stated, thermochemical data are for 298.15 K. Concerns over the harmful effects of chlorofluorocarbons on stratospheric ozone have motivated a search for new refrigerants. One such alternative is 2,2-dichloro-1,1,1-trifluoroethane (refrigerant 123). Younglove and McLinden published a compendium of thermophysical properties of this substance (J. Phys. Chem. Ref. Data 23, 7 (1994)), from which properties such as the Joule-Thomson coefficient $\mu$ can be computed. Compute $\mu$ at 1.00 bar and $50^\circ$C given that $(\partial H/\partial p)_T = -3.29 \times 10^3$ JMPa$^{-1}$ mol$^{-1}$ and $C_{p, \text{m}} = 110.0$ J K$^{-1}$ mol$^{-1}$.

[**Answer**] $\boxed{29.9}$

[**Intermediate Algebra**] To compute the Joule-Thomson coefficient $\mu$ at 1.00 bar and $50^\circ$C, we can use the following formula:

$$\mu = \left(\frac{\partial T}{\partial p}\right)_H$$

The Joule-Thomson coefficient can also be expressed in terms of the enthalpy and temperature derivatives as follows:

$$\mu = \frac{1}{C_{p,\text{m}}}\left(\frac{\partial H}{\partial p}\right)_T$$

Given:

$$\left(\frac{\partial H}{\partial p}\right)_T = -3.29 \times 10^3 \,\text{J}\,\text{MPa}^{-1}\,\text{mol}^{-1}$$

$$C_{p,\text{m}} = 110.0 \,\text{J}\,\text{K}^{-1}\,\text{mol}^{-1}$$

We can substitute these values into the formula to find $\mu$:

$$\mu = \frac{1}{110.0} \times (-3.29 \times 10^3) \,\text{K}\,\text{MPa}^{-1}$$

Let's calculate this using Python to ensure accuracy.

```
# Given values
dH_dp_T = -3.29e3  # J MPa$^{-1}$ mol$^{-1}$
Cp_m = 110.0  # J K$^{-1}$ mol$^{-1}$

# Calculate the Joule-Thomson coefficient
mu = dH_dp_T / Cp_m  # K MPa$^{-1}$
print(mu)
```

Output: $-29.90909090909091$

The Joule-Thomson coefficient $\mu$ at 1.00 bar and $50^\circ$C is:

$$\boxed{-29.91\,\text{K}\,\text{MPa}^{-1}}$$

Table 17: Chemistry example of self-verification in polymath learning.

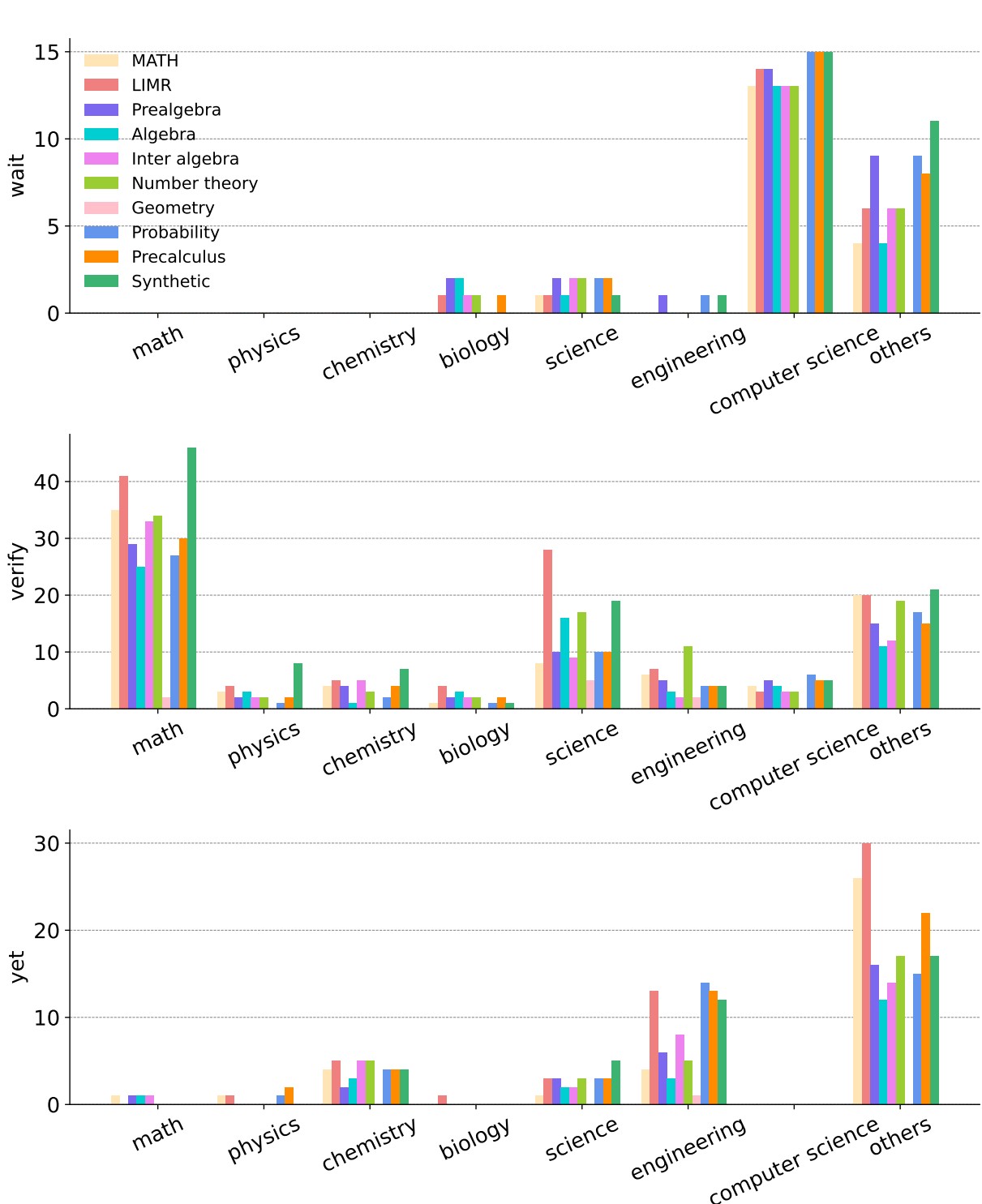

Figure 9: The verification patterns identified for 'wait', 'verify' and 'yet' in different subject domains. The 'wait' rates in computer science problems are highly attributed from terms in the question stems.

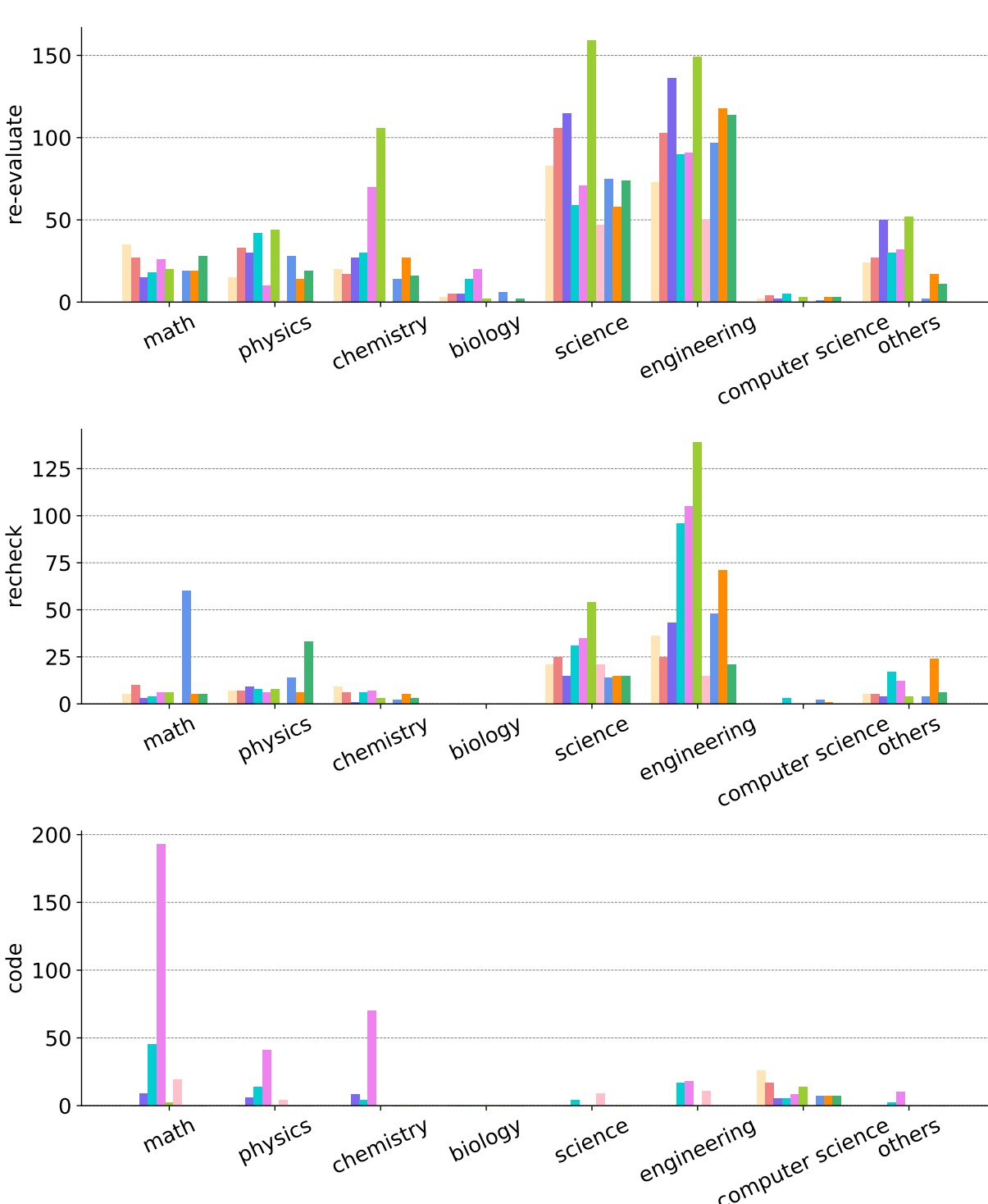

Figure 10: The verification patterns identified for 're-evaluate', 'recheck' and 'code' in different subject domains.

## N    OTHER POLYMATH SAMPLES

We list the other natural polymath samples from Table 18 to Table 24, and synthetic specialist samples from Table 25 to Table 29.

---

**Polymath Sample in Geometry**

**[Question]** A white cylindrical silo has a diameter of 30 feet and a height of 80 feet. A red stripe with a horizontal width of 3 feet is painted on the silo, as shown, making two complete revolutions around it. What is the area of the stripe in square feet?

[asy]
size(250);defaultpen(linewidth(0.8));
draw(ellipse(origin, 3, 1));
fill((3,0)–(3,2)–(-3,2)–(-3,0)–cycle, white);
draw((3,0)–(3,16)h-3,0)–(-3,16));
draw((0, 15)–(3, 12)h0, 16)–(3, 13));
filldraw(ellipse((0, 16), 3, 1), white, black);
draw((-3,11)–(3, 5)h-3,10)–(3, 4));
draw((-3,2)–(0,-1)h-3,1)–(-1,-0.89));
draw((0,-1)–(0,15), dashed);
draw((3,-2)–(3,-4)h-3,-2)–(-3,-4));
draw((-7,0)–(-5,0)h-7,16)–(-5,16));
draw((3,-3)–(-3,-3), Arrows(6));
draw((-6,0)–(-6,16), Arrows(6));
draw((-2,9)–(-1,9), Arrows(3));
label("3", (-1.375,9.05), dir(260), UnFill);
label("$A$", (0,15), N);
label("$B$", (0,-1), NE);
label("30", (0, -3), S);
label("80", (-6, 8), W);
[/asy]

**[Answer]** $\boxed{240}$

Table 18: Polymath sample in geometry.

---

**Polymath Sample (Probability)**

**[Question]** Bicycle license plates in Flatville each contain three letters. The first is chosen from the set $\{C, H, L, P, R\}$, the second from $\{A, I, O\}$, and the third from $\{D, M, N, T\}$.
When Flatville needed more license plates, they added two new letters. The new letters may both be added to one set or one letter may be added to one set and one to another set. What is the largest possible number of ADDITIONAL license plates that can be made by adding two letters?
**[Answer]** 40

Table 19: Polymath sample in counting and probability.

---

**Polymath Sample in Algebra**

**[Question]** A 100-gon $P_1$ is drawn in the Cartesian plane. The sum of the $x$-coordinates of the 100 vertices equals 2009. The midpoints of the sides of $P_1$ form a second 100-gon, $P_2$. Finally, the midpoints of the sides of $P_2$ form a third 100-gon, $P_3$. Find the sum of the $x$-coordinates of the vertices of $P_3$.

**[Answer]** $\boxed{2009}$

Table 20: Polymath sample in algebra.

## O   LARGE LANGUAGE MODEL USAGE

The large language model is employed to provide writing suggestions for polishing purposes.

**Polymath Sample in Intermediate Algebra**

[Question] Let $a$, $b$, $c$ be nonzero real numbers such that

$$\frac{a}{b} + \frac{b}{c} + \frac{c}{a} = 7 \quad \text{and} \quad \frac{b}{a} + \frac{c}{b} + \frac{a}{c} = 9.$$

Find

$$\frac{a^3}{b^3} + \frac{b^3}{c^3} + \frac{c^3}{a^3}.$$

[Answer] $\boxed{157}$

Table 21: Polymath sample in intermediate algebra.

**Polymath Sample in Precalculus**

[Question] For a certain value of $k$, the system

$$x + ky + 3z = 0,$$
$$3x + ky - 2z = 0,$$
$$2x + 4y - 3z = 0$$

has a solution where $x$, $y$, and $z$ are all nonzero. Find $\frac{xz}{y^2}$.

[Answer] $\boxed{10}$

Table 22: Polymath sample in precalculus.

**Polymath Sample in Number Theory**

[Question] The American Mathematics College is holding its orientation for incoming freshmen. The incoming freshman class contains fewer than 500 people. When the freshmen are told to line up in columns of 23, 22 people are in the last column. When the freshmen are told to line up in columns of 21, 14 people are in the last column. How many people are in the incoming freshman class?

[Answer] $\boxed{413}$

Table 23: Polymath Sample in Number Theory.

---

**Polymath Sample in Prealgebra**

[**Question**] A region is bounded by semicircular arcs constructed on the side of a square whose sides measure $2/\pi$, as shown. What is the perimeter of this region?

[asy]
path a=(10,0)..(5,5)–(5,-5)..cycle;
path b=(0,10)..(5,5)–(-5,5)..cycle;
path c=(-10,0)..(-5,5)–(-5,-5)..cycle;
path d=(0,-10)..(-5,-5)–(5,-5)..cycle;
path e=(5,5)–(5,-5)–(-5,-5)–(-5,5)–cycle;
fill(e,gray(0.6));
fill(a,gray(0.8));
fill(b,gray(0.8));
fill(c,gray(0.8));
fill(d,gray(0.8));
draw(a,linewidth(0.7));
draw(b,linewidth(0.7));
draw(c,linewidth(0.7));
draw(d,linewidth(0.7));
draw(e,linewidth(0.7));
[/asy]

[**Answer**] 4

Table 24: Polymath sample in prealgebra.

---

**Synthetic Specialist Sample in Precalculus**

[**Question**] A chemical factory discharges waste into a river at a rate of 500 cubic meters per day. The waste has an untreated pollutant concentration of 100 mg/L. The river has a flow rate of 24,500 cubic meters per day, and the waste mixes completely and instantly with the river flow. The pollutant degrades following first-order kinetics with a half-life of 5 days. The time for water to travel from the discharge point to a critical fish habitat is 5 days. To protect an endangered fish species (reflecting ethical considerations of intrinsic value in philosophy), the pollutant concentration at the habitat must not exceed 0.1 mg/L. If the concentration exceeds this limit, the probability of harm to the fish is 0.05 per mg/L of excess concentration. Due to legal regulations (incorporating law), if harm occurs, the factory is fined $10,000 per day. The factory can treat the waste to reduce the pollutant concentration before discharge. The treatment cost is $0.005 per cubic meter per mg/L reduction in concentration (incorporating economics and chemistry). Calculate the optimal initial concentration of pollutant in the treated waste (in mg/L) that minimizes the total daily cost (treatment cost plus expected fine), considering the interdisciplinary aspects of physics (degradation kinetics and flow), biology (fish protection), and mathematics (optimization).

[**Answer**] 10

Table 25: Synthetic Specialist Sample in Precalculus.

## Synthetic Specialist Sample in Number Theory

[**Question**] A pharmaceutical company develops a new drug for treating a specific condition. The drug has a biological half-life of 4 hours in the human body and a volume of distribution of 50 liters. Clinical trials determine that the minimum therapeutic concentration required for efficacy is 10 mg/L. The drug is administered as a single intravenous bolus dose at the beginning of each day to maintain concentrations at or above the therapeutic level for exactly 8 hours.

The manufacturing cost analysis shows that each 500 mg vial of the drug costs $2.50 to produce, and the entire vial must be used if opened. Regulatory requirements (reflecting legal and ethical considerations) mandate that the drug concentration must not drop below the therapeutic level during the 8-hour treatment period.

Considering the exponential decay of the drug concentration, calculate the required dose in milligrams. Then, determine the daily cost in dollars for administering this dose, providing the cost to one decimal place.

[**Answer**] $\boxed{10}$

Table 26: Synthetic Specialist Sample in Number Theory.

## Synthetic Specialist Sample in Geometry

[**Question**] A model cell membrane is represented by a cube-shaped vesicle with a side length of 10.0 nm. The membrane is a phospholipid bilayer made of two leaflets. Each phospholipid occupies exactly 1.50 $nm^2$ of surface area within a single leaflet. Assuming both leaflets cover the entire outer surface of the cube and ignoring membrane thickness and edge effects, how many phospholipid molecules are present in the bilayer?

[**Answer**] $\boxed{800}$

Table 27: Synthetic Specialist Sample in Geometry.

## Synthetic Specialist Sample in Probability

[**Question**] A molecular-biology lab purifies a circular plasmid that is exactly 3000 base pairs (bp) long.
• Each base pair contains two deoxyribonucleotides, and every nucleotide carries one phosphate ($PO_4^{3-}$) group.
• While the cells were growing, the medium contained the $\beta$-emitter $^{32}P$, so every phosphate in the plasmid is $^{32}P$-labelled.
• The radioactive isotope $^{32}P$ has a half-life of 14.0 days.

Immediately after purification, a tube that contains precisely 100 identical plasmid molecules shows an activity of 1024 disintegrations per minute (dpm). The tube is stored in a freezer, and—after an integral number of whole half-lives—the activity is measured again and found to be exactly 4 dpm.

To cross-check the number of phosphate groups, the plasmid DNA is then completely hydrolysed and the liberated phosphate is quantitatively precipitated as calcium phosphate according to
$$3\,Ca^{2+} + 2\,PO_4^{3-} \rightarrow Ca_3(PO_4)_2 \downarrow$$
The precipitation requires exactly $5.0 \times 10^{-7}$ mol of $Ca^{2+}$ ions, confirming the amount of DNA present (the stoichiometry is consistent and needs no further calculation here).

What is the number of $^{32}P$ half-lives that have elapsed between the two activity measurements?

[**Answer**] $\boxed{8}$

Table 28: Synthetic Specialist Sample in Probability.

---

### Synthetic Specialist Sample in Algebra

**[Question]** A plant that is heterozygous for two independent genes, G and H (genotype GgHh), is self-pollinated. Exactly 640 seeds are obtained.

**Biology:** Only seeds that are homozygous recessive for both genes (gghh) can synthesize the enzyme "Catalase-X".
**Chemistry:** Every gghh seed is placed in its own vial containing 0.0800 mol of hydrogen peroxide. Catalase-X instantly and completely decomposes the peroxide according to

$$2\,H_2O_2 \rightarrow 2\,H_2O + O_2(g).$$

Thus each qualifying vial releases pure $O_2$ gas.
**Physics:** The $O_2$ is dried, transferred to a 1.00 L rigid cylinder at 298 K, and all molecules are singly ionised $(O_2 \rightarrow O_2^+ + e^-)$. The ions are accelerated so that each has speed $v$ that makes its circular path radius exactly 0.0400 m in a uniform magnetic field $B = 1.00$ T perpendicular to their velocity $(m(O_2) = 32$ u, $1$ u $= 1.66 \times 10^{-27}$ kg, $q = 1.60 \times 10^{-19}$ C). Immediately after acceleration an electronic gate allows only the very first $O_2^+$ ion to continue; all later ions are blocked. That single ion has a 50% chance of striking a narrow slit that leads to a detector; otherwise nothing is recorded.

A vial is counted as a "success" if its lone transmitted ion hits the detector. All vials operate independently.

What is the expected number of "successes" after all 640 seeds have been processed?

**[Answer]** $\boxed{20}$

Table 29: Synthetic Specialist Sample in Algebra.

