# OpenReview forum: "One Sample to Rule Them All: Extreme Data Efficiency in RL Scaling"
_ICLR.cc/2026/Conference — ICLR 2026 Conference Withdrawn Submission_

### Official Review · Reviewer_wLoj · 2025-11-01

**Soundness:** 3
**Presentation:** 1
**Contribution:** 3
**Rating:** 0
**Confidence:** 2

**Summary:**

This paper proposes a method for generating a single math sample that, when post-trained with RL on, improves performance across other domains like biology, chemistry.

**Strengths:**

The method proposed by the others does increase performance across other domains as shown by Figure 1.

**Weaknesses:**

While the work of this paper is not my area of expertise, I find several issues with the current work. First of all, it is quite dense and hard to read. The authors keep referring to LIMR throughout the paper, and claim that they select the samples with the lower LIMR scores (line 162), but this method is never explained or introduced in the current manuscript, hence I am unsure what exactly this means.

Second, as far as I can understand, the authors use "salient skill identification" to construct their samples, but I am not sure how exactly they define the skills. Could the authors explain more here? It seems to be that there is an LLM employed to identify the skills needed to solve a particular problem, but how are these skills defined?

Thirdly, the experimental setup makes no mention of hyperparameter tuning. Have the authors chosen a fixed set of hyperparams without sweeping? If so, how was this decision made?

Lastly, I am not sure I understand exactly what the authors do once they generate the sample. Lines 204-205 mention that they use GRPO and "augment the polymath sample into the batch of 128" - I thought the point of this work was to train on a single sample, generated using their framework. If so, what are the other 127 samples in the batch?

**Questions:**

I have posed my questions in the weaknesses section.

**Details Of Ethics Concerns:**

Template does not follow ICLR template (margins are too narrow).

---

> ### Author Response · Authors · 2025-12-03
>
> We thank the reviewer for the response and would like to offer a few clarifications.
>
> **LIMR score computation** A brief introduction of how the LIMR score is computed and the qualitative meaning of the scores is included in line 157-164. We include further details of the LIMR computation in Appendix D. And more supporting experiments are included in Appendix E.
>
> **Salient skills** We rely on instructing Qwen2.5-72b-instruct in identifying the skills. The exact prompt to identify the salient skills is actually included in Table 7, Appendix A, and one example in Table 10, Appendix K for how such skills would look like. We consistently employ this setting throughout the experiments. We have updated the manuscript to better point this information in line 194.
>
> **Experimental setup** Our experiment setups largely follows [1], which is described in line 204. We do not conduct hyperparameter sweeping due to resource constraints.
>
> **Training batch configuration** The batch size is 128 among the experiments, all these 128 samples are duplicate of the same sample. This setup follows [1].
>
> Finally, our submission follows the official ICLR 2026 template.
>
> [1] Reinforcement Learning for Reasoning in Large Language Models with One Training Example. NeurIPS 2025

---

### Official Review · Reviewer_4tkZ · 2025-11-05

**Soundness:** 2
**Presentation:** 2
**Contribution:** 2
**Rating:** 4
**Confidence:** 4

**Summary:**

This paper shows that using one single training example (polymath sample or synthetic data), can improve Qwen2.5-7B on multiple domains beyond only domain, which shows better cross-domain transfer capability than common math dataset like MATH and LIMR.

**Strengths:**

1. The method is simple and effective; the results are good and reasonable. If RLVR on a single-domain (MATH) dataset brings limited gains or even causes forgetting in other domains, fewer-example RLVR may perform better for cross-domain improvement.
2. The writing and pipeline are clear, and the evaluation covers a comprehensive set of categories.
3. Also mention that training-free in-context learning can yield some improvement.

**Weaknesses:**

1. As mentioned in [1], the results should not be reported only on the Qwen2.5-7B model. Other one-shot RLVR–related works [2,3,4] also consider other models, like Llama-3 3B or 8B, and maybe other SFT models like OpenThinker3-1.5B. It’s not necessary to beat RLVR with MATH/LIMR on all models, but we should at least see significant improvement from using polymath/synthetic data. I also wonder whether the data transfer to other models, or whether we have to select data for each different model.
2. How does the selected data compare to the training data used in previous work [2,3,4]? I think it’s important to show the advantage of the data-construction pipeline in the paper by comparing with them.
3. How did you get the 1,500 random samples from SuperGPQA? I note that Qwen2.5-7B can get about 25–28% overall performance on the SuperGPQA benchmark, but only 15.7% in your report. Is this mainly affected by the prompt, the selected subset, or are they from the hard part? Similar issues may exist in GPQA Diamond. Although it’s fine to compare under the same evaluation pipeline, the gap is too large and needs explanation.

I think these questions are critical, and would like to increase my score if they are fixed.


[1] Shao, Rulin, et al. "Spurious rewards: Rethinking training signals in rlvr." arXiv preprint arXiv:2506.10947 (2025).
[2] Wang, Yubo, et al. "Unleashing the Reasoning Potential of Pre-trained LLMs by Critique Fine-Tuning on One Problem." arXiv preprint arXiv:2506.03295 (2025).
[3] Wang, Yiping, et al. "Reinforcement learning for reasoning in large language models with one training example." arXiv preprint arXiv:2504.20571 (2025).
[4] Gao Z, Chen L, Luo H, et al. One-shot entropy minimization[J]. arXiv preprint arXiv:2505.20282, 2025.

**Questions:**

Are you using the same compute for MATH/LIMR training and for one-shot training? Would a larger dataset require more compute to converge? Maybe we should include tables/figures showing accuracy vs. training steps to verify whether the results are converging.

---

> ### Author Response · Authors · 2025-12-04
>
> We thanks the reviewer for the efforts and acknowledge the effectiveness of our polymath learning.
>
> - **Results on additional models**: we include the results on additional models in point 3 of the general response. Polymath learning with the synthetic prime sample outperforms comprehensive learning in Qwen2.5-14b-base, and on several non-math domains in OctoThinker-8b-long-base.
> - **Comparison of prior 1-shot sample**: We include the result employing the sample from the mentioned work (denoted as pi_{1}), the results are included in point 5 of the general response where our polymath samples outperforms pi_{1}. Also note that pi_{1} does not come from the MATH dataset unlike the natural polymath samples.
> - **SuperGPQA evaluation setup**: The 1500 samples are randomly selected from the superGPQA set. We have also updated the manuscripts to include the performance on the full superGPQA set We would like to also clarify that the performance mentioned by the reviewer comes from 5-shot prompt for evaluation while we employ 0-shot prompt. Please refer to point 2 in the general response for details.
> - **Training dynamics** The computation between comprehensive learning and one-shot RLVR stays the same. We include the training dynamics over comprehensive learning and different polymath learning samples in Appendix H to show the more robust reasoning behavior in polymath learning.

---

### Official Review · Reviewer_xEUz · 2025-11-06

**Soundness:** 3
**Presentation:** 3
**Contribution:** 2
**Rating:** 4
**Confidence:** 2

**Summary:**

This paper introduces "polymath learning," demonstrating that a single, strategically selected math reasoning sample can improve LLM performance across diverse domains (physics, chemistry, biology) through reinforcement learning, often outperforming training on thousands of samples. The authors find that optimal samples exhibit salient algebra and precalculus skills, and that synthetically engineered problems integrating multidisciplinary knowledge achieve the best results, suggesting a shift from data scaling to precision "sample engineering" for more efficient reasoning enhancement.

**Strengths:**

1. Important research problem.
The paper proposes a meaningful problem to study: the data efficiency in current RL scaling for LLMs. The paper proposes an attemptive method to deal with the problem, which sheds some light on this important direction.

2. Clear Writing.
The writing is easy to follow, and the methods and experiments are clearly presented.

**Weaknesses:**

1. Unreasonably Low Math500 Performance after GRPO Training.
After GRPO fine-tuning, the Qwen2.5-7B model achieves only 37.2 accuracy on Math500, which is far below the expected score. This discrepancy raises concerns about the validity of the experiment results.

2. Lack of Robustness Verification for the Proposed Method
The paper does not provide sufficient evidence of the robustness and statistical reliability of the proposed sample selection method. A convincing validation would require multiple repeated experiments (e.g., 100 independent trials) and report the mean and variance of the final performance.

**Questions:**

1. Unsubstantiated Claim about Low-LIMR Preference
The authors argue that “high LIMR samples lead to over-specialization in mathematics” and therefore choose low-score (≈0.6) LIMR samples, yet provide no controlled experiments to establish a causal relationship or robustness.

2. Limited Model Diversity and Poor External Validity
Training and evaluation are both performed exclusively on Qwen2.5-7B-Base. The study does not examine how results generalize across different model sizes or architectures. This narrow setup restricts the external validity and robustness of the conclusions.

---

> ### Author Response · Authors · 2025-12-04
>
> We thank the reviewer for the suggestions. For the questions
>
> - **Low MATH500 performance**: The result in Table only includes performance on the **Math domain**, which includes MATH500, the dataset-wise result is actually included in Table 9, Appendix C, where both comprehensive learning and polymath learning is able to achieve above 70 on MATH500.
> - **Robustness of the results**: we have included independent runs over our major experiments, where polymath learning on the synthetic sample consistently outperforms comprehensive learning, please refer to point 4 in general response and results in Appendix F.
> - **Evidence for low-LIMR preference**: we have updated LIMR score selection results in Appendix E of the manuscript.
> - **Diversity of performance**: we include the results from additional models in point 3 in the general response.

---

### Official Review · Reviewer_yKH5 · 2025-11-12

**Soundness:** 1
**Presentation:** 2
**Contribution:** 1
**Rating:** 2
**Confidence:** 4

**Summary:**

This paper investigates whether reinforcement learning on a single training sample can enhance reasoning in large language models across multiple domains. The authors introduce Polymath Learning, a framework for one-shot RL training designed to extract cross-domain reasoning improvements from a single carefully chosen math problem.

**Strengths:**

1. This paper shows that RL can be extremely efficient on one single example.
2. This paper demonstrates RL on a well-designed cross-domain reasoning problem can improve model's reasoning performance across domains.

**Weaknesses:**

1. The evaluation is done on a randomly sampled subset (100 problems for each subject) from multiple benchmarks, this seems not a standard evaluation and makes it hard to compare the experimental results with prior works.
2. Only one Qwen base model is tested, it's unclear how the method generalizes to other model families.
3. The sample selection relies on the LIMR score, which actually requires a complete RL training on the full dataset. Therefore, the proposed method is not as efficient as it claims to be "one-shot RL".

**Questions:**

1. Can the authors provide a principled explanation on why one-shot RL can be better than RL on the whole dataset?
2. Can the authors include the reward curve across training steps (e.g., mean reward) to illustrate how the model’s behavior evolves during one-shot training?
3. Are the results reported in Table 3 statistically significant? For example, are they at least beyond the 2-sigma or 95% confidence level?
4. Can the authors clarify what insight Figure 2 is intended to convey?
5. Why does RL on a single training example not lead to severe overfitting? Recent works (e.g., [1], [2]) show that reinforcing correct samples can reduce output entropy and lead to overconfidence, and that RL often refines a model’s prior knowledge rather than improving its intrinsic capabilities. Could the observed one-shot improvements arise from implicit refinement of pre-existing knowledge rather than genuine reasoning enhancement?

[1] The Surprising Effectiveness of Negative Reinforcement in LLM Reasoning. NeurIPS 2025

[2] Does Reinforcement Learning Really Incentivize Reasoning Capacity in LLMs Beyond the Base Model? NeurIPS 2025

---

> ### Author Response · Authors · 2025-12-04
>
> We thank the review for the response. For the question of the reviewer
>
> - **Sampled benchmarks**: we have included the results of full evaluation set of MMLU-Pro and SuperGPQA in point 1 of the general response, where polymath learning on the synthetic prime sample outperforms comprehensive learning.
> - **Results on additional models**: we include the results on additional models in point 3 of the general response. Polymath learning with the synthetic prime sample outperforms comprehensive learning in Qwen2.5-14b-base, and on several non-math domains in OctoThinker-8b-long-base.
> - **Computational efficiency**: while the natural polymath samples relies on LIMR score, the synthetic sample does not require prior RL training.
> - **Explanation of the effectiveness on one-shot RL training**: while [1] shows that 1-shot RL could achieve better math reasoning abilities, [2] provides more theoretical efforts and demonstrates that for strong model generator, data pruning is more effective than data volume.
> - **Training dynamics**: We have updated the training dynamics of comprehensive learning and signature polymath samples in Appendix H.
> - **Result robustness**: We have updated the results of independent runs in Appendix F, and in point 4 of the general response.
> - **Meaning of Figure 2**: The results on Figure 2 shows the skewed multidiscipline reasoning ability of the base model—it is comparatively strong on math but noticeably weaker on other subjects. Besides, providing an in-context polymath learning sample improves performance more than comprehensive learning on 8k MATH samples across a broad range of subjects. Moreover, the improvement of polymath learning is especially pronounced in subjects that are more distant from math, suggesting that polymath learning encourages more transferable reasoning.
> - **Implicit refinement**: Training on single-sample is easier for the emergence of “reward saturation” behavior, the ablation in [1] suggests the critical role of the policy gradient loss while entropy loss provides further improvement.
>
> References
>
> [1] Reinforcement Learning for Reasoning in Large Language Models with One Training Example. NeurIPS 2025
>
> [2] Why Less is More (Sometimes): A Theory of Data Curation. arXiv

---

### Author Response · Authors · 2025-12-04
**General Response to Reviewers and ACs**

We thank all reviewers for their constructive feedback. Below we summarize the common questions and provide clarifications that we will incorporate into the revised manuscript.

In this work, we propose polymath learning, a different learning paradigm from learning from thousands of training samples (comprehensive learning). In polymath learning, the model learns from a **single carefully selected or engineered “polymath sample”**. Our results show that carefully selected or engineered polymath samples are able to outperform comprehensive learning on multidiscipline reasoning problems.

1. **The comparison with full MMLU-Pro and SuperGPQA** (reviewer yKH5, 4tKz).
    - To maintain subject-balanced comparison, we include a sampled set from MMLU-Pro and SuperGPQA in major experiments. Here, The performance differences across full vs. sampled sets are shown below, where **polymath learning (Synthetic Prime) yields higher multidisciplinary performance.**

| Sample | MMLU-Pro | SuperGPQA |
| --- | --- | --- |
| 0-shot | 30.3 / 4.7 | 16.8 / 15.7 |
| MATH | 31.7 / 22.5 | 16.6 / 10.3 |
| Synthetic Prime | **37.6 / 38.4** | **21.7 / 20.5** |

*a/b denote results on the full evaluation set (left) and the sampled subset (right)*

2. **Qwen2.5-7b-base performance in benchmarks** (reviewer xEUz, 4tkZ).
    - Our experiments employs a **zero-shot prompt**. The previously mentioned higher numbers in the superGPQA work employs **5-shot prompt** for Qwen2.5-7b-base, which is not directly comparable to our setting.
    - Note that the official prompts of MMLU-Pro and SuperGPQA are tailored for **multiple-choice** problems because of the benchmark nature, while our prompt needs to incorporate both open-ended and multiple-choice reasoning problems.
    - The Math results in Table 3 correspond to the **“Math” domain** (clarified in line 246, including math problems in all benchmarks), not the **MATH dataset**. The full **MATH500** results (Table 9, Appendix C) are higher than initially inferred.


3. **Additional model results of polymath learning** (reviewer yKH5, 4tkZ)

- We have conducted experiments over several other models (Qwen2.5-14b-base, llama3.1-8b-instruct, OctoThinker-8b-long-base), here is our major findings
    - Synthetic Prime consistently improves performance over 0-shot across models.
    - For stronger models like Qwen2.5-14b-base, polymath learning on the synthetic prime sample demonstrates stronger performance than comprehensive learning on the full training set. However, such effectiveness does not extend to weaker models like Llama3.1-8b-instruct, a trend consistent with prior observations in 1-shot RLVR [1] and theoretical results on data selection [2], which suggests that **data pruning is more effective than data volume when the model generator is strong** and explains the effectiveness of polymath learning on Qwen2.5-7b-base and Qwen2.5-14b-base.
    - For Octothinker-8b-long-base [3], which employs mid-training from Llama, polymath learning demonstrates superior effectiveness on several **non-math** domains over comprehensive learning such as Computer Science, Engineering, and Others, which aligns our findings about math-distant subjects in line 360.

- Qwen2.5-14b-base

| Sample | Math | Physics | Chemistry | Biology | Science | Engineering | CS | Other | Avg |
| --- | --- | --- | --- | --- | --- | --- | --- | --- | --- |
| 0-shot | 37.7 | 26.2 | 22.2 | 28.1 | 41.2 | 39.0 | 20.8 | 14.3 | 28.7 |
| MATH-full | 42.7 | 26.4 | 20.5 | 44.7 | 49.5 | **64.4** | 22.3 | 15.6 | 35.8 |
| synthetic prime | **44.0** | **32.7** | **22.7** | **42.3** | **56.7** | 58.5 | **31.0** | **20.6** | **38.6** |

- Llama3.1-8b-instruct


    | Sample | Math | Physics | Chemistry | Biology | Science | Engineering | CS | Other | Avg |
    | --- | --- | --- | --- | --- | --- | --- | --- | --- | --- |
    | 0-shot | 22.4 | 5.4 | 5.5 | 8.8 | 8.1 | 18.8 | 2.0 | 1.7 | 9.1 |
    | MATH-full | **26.5** | **9.8** | **9.6** | **28.4** | **34.3** | **50.4** | **9.6** | **6.6** | **21.9** |
    | Synthetic Prime | 22.2 | 7.9 | 7.4 | 7.1 | 9.1 | 20.5 | 2.4 | 1.5 | 9.8 |

- OctoThinker-8b-long-base (8k length)

| Sample | Math | Physics | Chemistry | Biology | Science | Engineering | CS | Other | Avg |
| --- | --- | --- | --- | --- | --- | --- | --- | --- | --- |
| 0-shot | 5.9 | 0.5 | 0.4 | 0.1 | 0.0 | 0.0 | 0.0 | 0.0 | 0.9 |
| MATH-full | **37.9** | **18.7** | **17.1** | **35.6** | **58.6** | 50.4 | 16.0 | 11.4 | **30.7** |
| Synthetic Prime | 10.0 | 16.5 | 11.8 | 24.9 | 37.4 | **56.4** | **20.6** | **13.8** | 23.9 |

---

> ### Author Response · Authors · 2025-12-04
>
> 4. **Stability of Polymath Learning** (reviewer yKH5, xEUz)
>     - We have conclude 3 independent runs of MATH full and polymath learning on the synthetic prime sample. The results show that **polymath learning on the synthetic prime sample consistently outperforms comprehensive learning on the full MATH training set**. Please refer to Appendix F for the results and Appendix H for training dynamics information in the updated manuscript.
>
> 5. **Comparison with Other 1-shot Sample** (reviewer 4tkZ)
>
> Our natural polymath sample are selected from the **MATH training set**. Here we also include pi_{1} sample employed in prior works [1, 5, 6]. Note that  pi_{1} is selected from **DeepScaleR [4], a dataset curated from challenging math competition problem**.
>
> -  **Performance comparison** The results on Qwen2.5-7b-base and Qwen2.5-14b base show that **the synthetic prime sample consistently outperform pi_{1}**. Moreover, **most our natural polymath samples outperforms pi_{1} even though they come from a less challenging sample pool** (please refer to Table 3 in the manuscript).
> - **Training dynamics** The training dynamics further indicate that our polymath samples provide stronger reasoning incentivization than \pi_{1} (see Appendix H).
>     -  Qwen2.5-7b-base
>
> | Sample | Math | Physics | Chemistry | Biology | Science | Engineering | CS | Other | Avg |
> | --- | --- | --- | --- | --- | --- | --- | --- | --- | --- |
> | 0-shot | 20.4 | 4.4 | 4.4 | 5.1 | 0.0 | 3.7 | 3.3 | 9.6 | 6.4 |
> | MATH-full | 37.2 | 12.8 | 10.0 | 23.5 | 25.8 | 31.4 | 8.6 | 6.5 | 19.5 |
> | pi_1 | 35.5 | 14.3 | 11.3 | 28.4 | 35.1 | 44.1 | 13.8 | 10.4 | 24.1 |
> | synthetic prime | **38.3** | **20.6** | **15.7** | **32.4** | **48.5** | **54.2** | **20.8** | **15.6** | **30.8** |
>
> -  Qwen2.5-14b-base
>
>
>     | Sample | Math | Physics | Chemistry | Biology | Science | Engineering | CS | Other | Avg |
>     | --- | --- | --- | --- | --- | --- | --- | --- | --- | --- |
>     | 0-shot | 37.7 | 26.2 | 22.2 | 28.1 | 41.2 | 39.0 | 20.8 | 14.3 | 28.7 |
>     | MATH-full | 42.7 | 26.4 | 20.5 | 44.7 | 49.5 | **64.4** | 22.3 | 15.6 | 35.8 |
>     | pi_1 | 40.4 | 27.6 | 20.0 | 39.4 | 51.5 | 57.6 | 22.1 | 17.1 | 34.5 |
>     | synthetic prime | **44.0** | **32.7** | **22.7** | **42.3** | **56.7** | 58.5 | **31.0** | **20.6** | **38.6** |
>
> 6. **Major Manuscript Updates**
> -  LIMR score basics: Appendix D.
> -  Sample preference results based on LIMR scores: Appendix E.
> -  Robustness of polymath learning: Appendix F.
> -  Comparison with full set of MMLU-Pro and SuperGPQA: Appendix G
> - Training dynamics between comprehensive learning and different polymath samples: Appendix H
> - Comparison with pi_{1}: Appendix I
>
> We thank again for the efforts of the reviewer and AC.
>
> **References**
>
> [1] Reinforcement Learning for Reasoning in Large Language Models with One Training Example. NeurIPS 2025
>
> [2] Why Less is More (Sometimes): A Theory of Data Curation. arXiv
>
> [3] OctoThinker: Mid-training Incentivizes Reinforcement Learning Scaling. arXiv
>
> [4] DeepScaleR: Effective RL Scaling of Reasoning Models via Iterative Context Lengthening. arXiv
>
> [5] Unleashing the Reasoning Potential of Pre-trained LLMs by Critique Fine-Tuning on One Problem. arXiv
>
> [6] One-shot entropy minimization. arXiv

---

### Note · Authors · 2026-01-06

**Comment:**

We thank for the efforts of the reviewers and the AC. We decide to withdraw our work from submission.

**Withdrawal Confirmation:**

I have read and agree with the venue's withdrawal policy on behalf of myself and my co-authors.